# Impacts of land use/cover change and reforestation on summer rainfall for the Yangtze River Basin

Wei Li[1,2], Lu Li[3], Jie Chen[1,2], Qian Lin[1,2], Hua Chen[1,2]

[1]State Key Laboratory of Water Resources and Hydropower Engineering Science, Wuhan University, Wuhan, 430072, China
[2]Hubei Key Laboratory of Water System Science for Sponge City Construction, Wuhan University, Wuhan, 430072, China
[3]NORCE Norwegian Research Centre, Bjerknes Centre for Climate Research, Bergen, Jahnebakken 5, 5008, Bergen, Norway

*Corresponding to*: Jie Chen (jiechen@whu.edu.cn)

**Abstract.** Land use and cover have been significantly changed all around the world during the last decade. In particular, the Grain for Green (GG) has resulted in significant changes in regional land use and cover, especially in China. The land use and cover change (LULCC) may lead to changes in regional climate. In this study, we take the Yangtze river basin as a case study and analyse the impacts of LULCC and reforestation on summer rainfall amounts and extremes based on the Weather Research and Forecasting model. Firstly, two observed land use and cover scenarios (1990 and 2010) were chosen to investigate the impacts of LULCC on the summer rainfall during the last decade. Secondly, two hypothetical reforestation scenarios (i.e., scenarios of 20% and 50% cropland changed to be forest) were taken based on the control year of 2010 to test the sensitivity of summer rainfall (amounts and extremes) to reforestation. The results showed that average summer rainfall and extreme summer daily rainfall decreased in the Yangtze River basin between 1990 and 2010 due to LULCC. Reforestation could increase summer rainfall amount and extremes, and the effects were more pronounced in the populated area than over the whole basin. Moreover, the effects of reforestation were influenced by the reforestation proportion. In addition, the summer rainfall increased less conversely with the transform proportion of cropland to forest increased from 20% to 50%. By analysing the changes in water vapor mixing ratio, upward moisture flux and 10m wind, it suggested that this result might be caused by the horizontal transportation processes of moisture. Although a comprehensive assessment of the impacts of LULCC on summer rainfall amounts and extremes was conducted, further studies are needed to investigate the uncertainty better.

## 1 Introduction

Human activities intensified land use and land cover change (LULCC) all around the world. With the human population increasing, more than one-third of global natural land uses were altered by human activities during the past three centuries (Hurtt et al., 2006, 2011). The land surface was the lower boundary of atmospheric motion. Thus, LULCC could influence climate through various geophysical processes, such as the water and heat flux between land surface and atmosphere, surface wind speed, and boundary layer turbulence. LULCC could affect regional climate significantly, which had become a broad consensus as many studies proved this. For instance, Pitman et al. (2012) found that many of the temperature indices showed

locally strong and statistically significant responses to LULCC, such as that commonly 30-50% of the continental surfaces of the tropics and Northern and Southern Hemispheres were affected statistically significantly by LULCC. Wen et al. (2013) also found that land use changes in China could contribute to the warmest day temperature increases. Furthermore, Yu et al. (2020) found that the recent greening in China inferred a country-averaged surface cooling of 0.11 ℃. Lin et al. (2020) showed that the urbanization tended to weak extreme rainfall events in urban agglomerations over coastal regions and intensify the

influences on those in central/western China.

China is experiencing significant land use changes due to human activities, especially for the high-population-density Yangtze River basin (YRB). The Yangtze River is the longest river in Asia and the third-longest globally, with a length of over 6300 km. The YRB is the largest basin in China, which supports 34% of the national population and contributes 41.1% of China's gross domestic products (Zhang et al., 2014). Considering agricultural activities, urbanization, and dam construction, LULCC

is quite significant in this basin (Liu et al., 2003; Zhang et al., 2009; Shen et al., 2019; Lin et al., 2021). Moreover, China launched the Grain for Green (GG) to expand forestland in 1999, aiming to reduce soil erosion and alleviate poverty (Robbins and Harrell, 2014; Li et al., 2020). From 1999 to 2008, forest coverage, reported as a percentage of the country's total land area, increased from 16.55 % to 20.36 %, adding 41.6 million ha of forest (Trac et al., 2013). By 2013, China government had invested over 320 billion RMB in afforesting over 29 million hectares (Zinda et al., 2017). The GG focused on increasing

forest cover through cropland conversion and afforestation and reforestation of barren hillsides. Sloping cropland was a core target of the program, which was blamed for 65% of the 2 to 4 billion tons of silts released into the Yangtze and middle and upper reaches of the Yellow River each year (Bennett et al., 2011). Because of the GG, there was also a trend of LULCC in the YRB that returning cropland to forest. All the LULCC in the YRB changed the regional climate during the past few decades. For example, Cui et al. (2012) found that reforestation could increase evapotranspiration and decrease water yield at the forest

stand level in the upper reach of the YRB. Liu et al. (2013) displayed that reforestation in the upstream of the YRB increased annual evapotranspiration, leading to reductions in surface flow and baseflow. Besides, Hu et al. (2015) found that LULCC in eastern China caused a decrease in rainfall over the lower reaches of the YRB of approximately 3% in the summer from the 1980s to the 2000s. Zhang et al. (2017) showed that the temperature decreased by 0.2-0.4℃ in the midstream and downstream of the YRB in spring, autumn and winter, and the seasonal rainfall also decreased from the 1980s to the 2000s due to LULCC.

Furthermore, Feng et al. (2018) showed that the land surface temperature over the Taihu Lake Basin, which was located in the lower reaches of the YRB, has been increasing since 1996 caused by local urbanization.

The YRB plays a vital role in ecosystem protection and economic development for the whole country. However, the YRB suffered from flooding frequently during the past decades. Summer rainfall from June to August is the leading cause of summer flooding in the YRB, which largely influences the lives of local people. Thus, it is crucial to better understand the impacts of

LULCC on summer rainfall in the YRB, especially the effects of the GG reforestation program. Although many studies estimated the impacts of LULCC on rainfall in the YRB, it should be noted that most of the previous studies only focused on the midstream and downstream of the YRB. Moreover, the sensitivity of summer rainfall to reforestation in the YRB was rarely investigated. And few previous studies discussed the potential physical mechanisms linked to the changes in summer rainfall under reforestation. To investigate the impacts of LULCC, especially the reforestation on rainfall, is of great

importance for the economic and ecological development of the YRB as well as for China. There is an urgent need considering the Yangtze River Coordinated Protection Strategy proposed by the Chinese government in 2016, aiming to prioritise ecology and green development, promote well-coordinated environmental conservation, and avoid excessive development.

Therefore, this study took the YRB as a case study and investigated the impact of LULCC and reforestation on summer rainfall and extreme hazards (from June to August). More specifically, two observed LULCC scenarios were chosen to study the

impacts of observed LULCC on summer rainfall, including both amounts and extremes, while two hypothetical reforestation scenarios were taken to quantitively assess the impacts of reforestation on summer rainfall (amount and extremes) under different reforestation proportions. The differences in summer rainfall between the four land use scenarios (two observed and two hypothetical ones) were applied and investigated based on the Weather Research and Forecasting (WRF) model. The major objectives of this study were to: (1) estimate the impacts of LULCC and reforestation on summer rainfall (amount and

extremes) in the YRB; and (2) investigate how the proportion of reforestation affects summer rainfall (amount and extremes) in the YRB.

To better understand the impacts of LULCC and reforestation on summer rainfall, the performance of WRF-simulated rainfall was first evaluated in section 4.1. Then, the changes in summer rainfall between the 1990 scenario and 2010 scenario were analysed to investigate the impacts of observed LULCC on summer rainfall in section 4.2. In sections 4.3.1 and 4.3.2, the

impacts of reforestation on summer rainfall were analysed based on the spatial changes and area average changes, respectively. Moreover, in section 4.3.3 and 4.3.4, the impacts of reforestation on some other climate variables related to the rainfall were also investigated. These climate variables contained the latent heat flux (LHF), sensible heat flux (SHF), planetary boundary layer height (PBLH), 2m air temperature, 2m relative humidity, and 2m water vapor mixing ratio. The analyses of these variables aimed to explain the potential mechanisms of the changes in summer rainfall under reforestation. The discussions

and conclusions are given at the end. Our results will contribute to a better understanding of regional climate characteristics (summer rainfall and extremes) under the impacts of LULCC and the reforestation program in the YRB and provide a knowledge base for ecological reconstruction programs in the future.

## 2 Study area and data

### 2.1 Study area

This study focuses on YRB (Fig. 1), which has a total area of ~$1.8 \times 10^6$ km$^2$ (Wang et al., 2018). The YRB is located between 24°-35°N and 90°-122°E, spanning from the eastern Tibetan Plateau to the East China Sea and crossing 19 provinces in China. The upper, middle, and lower reaches of the YRB cover $1.0 \times 10^6$ km$^2$, $6.8 \times 10^5$ km$^2$, and $1.2 \times 10^5$ km$^2$, respectively (Zhang et al., 2014). LULCC in the YRB was quite significant during the past few decades. The main types of LULCC including urbanization which leads to the conversion of cropland to the urban area in the middle and lower reaches (Liu et al., 2010, 2012; Gao et al., 2012), degradation of grassland caused by overgrazing in the headwater region (Gao et al., 2009;2010), and reforestation and afforestation as a result of the implementation of the GG (Liu et al., 2010; Li et al., 2014). The upper reaches of the YRB belong to a high-cold climate zone, whereas the middle and lower reaches belong to subtropical and temperate climate zones (Zhang et al., 2019a). The whole YRB is sensitive and vulnerable to climate change (Fang et al., 2010). The average air temperature ranges from 9 to 18 °C, and the average annual rainfall ranges from 692 to 1611 mm (Zhang et al., 100 2019a). The flash flooding in the Yangtze River basin is often caused by continuous rainfall that lasts for a few days, as it is a big basin (Chen et al., 2020; Nanding et al., 2020). A few hours of high-intensity rainfall do not cause severe flooding due to cascade reservoirs' construction along the river. Because of relatively good water and temperature conditions, vegetation productivity is generally high in this area. However, human activities are intensifying LULCC in the YRB from the upper reaches to the lower reaches (Sun et al., 2016), which will gradually change the local climate and influence agriculture 105 production.

### 2.2 Data

This study used WRF simulations to investigate the impacts of LULCC on summer rainfall. The WRF model with the Advanced Research WRF dynamics solver version 3.9.1 was used (Skamarock et al., 2008). The WRF model was a flexible, state-of-the-art, non-hydrostatic, mesoscale numerical weather prediction, and atmospheric simulation system (Wagner et al., 110 2016). The lateral boundaries of the WRF model were forced with the 0.5° ERA-Interim reanalysis (Berrisford et al., 2011). The output interval of the WRF model was one day. The output variables of the WRF model contained rainfall, LHF, SHF, PBLH, 2m air temperature, 2m relative humidity, and 2m water vapor mixing ratio were used in this study to analyse the changes in rainfall under LULCC and the potential physical mechanisms.

Besides, the observed rainfall and temperature from 171 stations in the YRB were used for model validation (Fig. 1). The 115 observed data were quality controlled and provided by the China Meteorological Data Sharing Service System. In our study, the observed data from stations were interpolated to model grids by the Inverse Distance Weight (IDW) interpolation method.

Meanwhile, to better validate the model performance, the ERA5 data were used as the benchmark to validate some thermodynamics variables as we did not have gauged observations for these variables. The spatial resolution of ERA5 data is 0.25°. In this study, the ERA5 data were interpolated to model grids by using the bilinear interpolation method. Moreover, to better understand the impacts of LULCC and reforestation on human lives, the 2010 Grid Population Dataset of China developed by Fu et al. (2014) was used. This dataset was developed based on remote sensing derived land use types and statistical population data. The spatial resolution of this dataset is 1km. This dataset was also interpolated to model grids using the IDW method.

In addition, the 1990 and 2010 land use data of the YRB were derived from the Landsat thematic mapper (TM) digital images (http://www.dsac.cn/ServiceCase/Detail/174574). It was interpreted based on the geometric shape, texture features, spatial distribution of the ground objects, and the spectral characteristics of images. Moreover, the outdoor survey and random sample check were also taken to enhance the accuracy of land use data. The land use changes were included in the WRF modelling by modifying the static geographical data, which further changed the simulation of subprocesses such as the vegetation phenology, canopy stomatal resistance, runoff, and groundwater in the land surface model Noah-MP (Li et al., 2018). Many parameters were used in Noah-MP to describe the characteristics of different land use types, such as albedo, HVT (Top of canopy), LAI (Monthly leaf area index), and VCMX25 (Maximum rate of carboxylation at 25 °C). When the land use changed, these parameters changed accordingly, which finally led to the changes in substance and energy exchanges between the atmosphere and land surface. In the study, we used the U.S. Geological Survey (USGS) land cover with 30s resolution (~ 1km resolution; "landuse_30s_with_lakes") in the WRF Preprocessing System (WPS). The new land use data of 1990 and 2010 derived from the Landsat TM digital images at 1km resolution, was then used to replace the USGS land cover data in the WRF simulation in YRB. As the resolutions of the outer and inner WRF domain were set as 75km and 15km, respectively, the post-processed land use data was resampled from 1km to 75km and 15km by using the WPS. The dominant land use categories in model grids were then used for the Noah-MP model to reflect the intended land use changes correctly. The land use categories of the 1990 and 2010 land use data from Landsat TM digital images were defined by Liu et al. (2002, 2005), which were commonly used in China; while, the USGS data for WRF modelling has 24 land use categories (including lake). Thus, we used the method of land use type conversions based on the study of Hu et al. (2015). According to this method, the four classes of land use in the Liu's category from Landsat TM digital images, including the Forest (Liu code 21), Shrub (Liu code 22), Sparse woodland (Liu code 23), and Cut over land (Liu code 24), were converted to four classes of USGS land use category, including the Deciduous broadleaf forest (USGS code 11), Shrubland (USGS 45 code 8), Savanna (USGS code 10), and Savanna (USGS code 10), respectively.

## 3 Methods

### 3.1 WRF Model configuration

The WRF model was set up with two nested domains in this study (Fig. 2). The resolutions of the outer and inner domain were 75 km (95 × 82 grids) and 15 km (236 × 161 grids), respectively. The model was set up with 32 vertical levels, and the top was at 50 hPa in all domains. The simulated period was 11 years from 2000 to 2010, with the first year taken as spin-up time. The initial and lateral boundary conditions were taken from the 0.5° ERA-Interim reanalysis data set. The time step was 90 s in both domains.

The choices of the microphysical scheme and cumulus parameterization are important for rainfall simulations (Li et al., 2017). According to previous studies in China (Hu et al., 2015; Zhang et al., 2019b; Feng et al., 2012; Xue et al., 2017), three microphysical schemes, i.e., Purdue Lin Scheme (Lin) (Chen and Sun, 2002), WRF Single-moment 5-class Scheme (WSM5) (Hong et al., 2004) and Eta (Ferrier) Scheme (Ferrier) (Rogers et al., 2001), and two cumulus parameterizations, i.e., Kain–Fritsch Scheme (KFN) (Kain, 2004) and Grell–Devenyi Ensemble Scheme (GD) (Grell and Dévényi, 2002), were chosen to validate the WRF model. Five parameterization schemes combinations (i.e., Lin-KFN, WSM5-KFN, Ferrier-KFN, Lin-GD and WSM5-GD) were then used to simulate the rainfall and temperature in the YRB during the 2005 summer (from June to August), as there were several rainstorm events in 2005 summer for this basin. The most suitable parameterization schemes were chosen by comparing the performance of these five combinations in simulating these rainstorm events. The domain setting was same as the whole experiment which can be seen in Fig. 2. Finally, the Lin and GD were set as microphysical scheme and cumulus parameterization, respectively.

Besides, the Yonsei University scheme was used for planetary boundary layer (Hong et al., 2006); the Dudhia scheme for shortwave radiation (Dudhia, 1988); the RRTM scheme for longwave radiation (Mlawer et al., 1997), and the Noah–MP scheme for the land surface model (Niu et al., 2011; Yang et al., 2011).

### 3.2 The observed land use scenarios and hypothetical reforestation scenarios

The 1990 and 2010 land use scenarios were chosen to estimate the impacts of observed LULCC on summer rainfall amount and extremes in this study (Fig. 3a and 3b). From 1990 to 2010, the YRB suffered significant LULCC. In this period, the main LULCC in the YRB was urbanization and reforestation and the constructions of dams (Liu et al., 2003; Zhang et al., 2009; Shen et al., 2019). Furthermore, to investigate the impacts of reforestation due to the GG, we randomly changed 20% and 50% of the cropland to be forest based on the observed land use scenario of 2010 (Fig. 3c and 3d). These two reforestation scenarios were independently produced using random sampling and can be considered as two extreme cases in the progress of GG for the future. The hypothetical reforestation scenarios (named with 20% scenario and 50% scenario) were used in the study as

well as the observed land use in 1990 and 2010 (named with 1990 scenario and 2010 scenario). When we changed croplands to forests, the proportions of each type of croplands (forests) occupied in total croplands (forests) were kept fixed. Moreover, Table 1 displayed the percentages of land use classes under four scenarios, while Table S1 displayed the percentages of land cover under four scenarios after resampled to 15km.

## 4 Results

### 4.1 WRF Model validation

Figure 4 displays the spatial distributions of biases in the average summer rainfall and extreme summer daily rainfall (the 90th and 99th percentile of summer daily rainfall) simulated by WRF relative to observation and the qq-plot of observed rainfall versus simulated rainfall. From Fig. 4a, it can be seen that the biases of WRF-simulated average summer rainfall range from -120% to 200%. The positive biases are mainly observed in the transaction region between Sichuan Basin and Tibet plateau,

with the maximum positive biases in the front zone of the Tibet plateau where the altitudes shift from low to high rapidly. The negative biases are mainly observed in the southeastern YRB, which were also found in other studies (Zhang et al., 2017). Figures 4b and 4c present the biases of 90th and 99th percentiles of summer daily rainfall simulated by WRF relative to observations, respectively. The 90th and 99th percentiles are average values over 10 years. The biases of 90th and 99th percentile of summer daily rainfall have almost identical spatial distributions and vary from -80% to 200%. The positive biases

are mainly observed in the upstream area where the altitudes are higher than 1200 m, while the negative biases are mainly observed in the midstream and downstream areas with the maximum negative biases located in the south-eastern YRB. The qq-plot of observed basin-averaged rainfall versus simulated basin-averaged rainfall in Fig. 4d shows that the distribution of basin-averaged rainfall simulated by WRF is linearly correlated with that of observation.

Figure 5a presents the basin-averaged summer rainfall processes (from June to August) of observed, ERA5 and WRF-

simulated rainfall (2010 scenario). The summer rainfall processes are the multiyear-averaged results of 10-year data from 2001 to 2010. It can be seen that ERA5 rainfall is overestimated comparing with the observed rainfall. The rainfall simulated by WRF falls within the spread between the observation and ERA5 rainfall at the beginning of summer and then become smaller than observation. Figure 5b presents the probability distribution functions (PDFs) of observed, ERA5 and WRF-simulated summer rainfall. The PDF of WRF-simulated rainfall is larger than that of observation when rainfall is smaller than 3 mm/day.

The PDF of ERA5 rainfall is larger than that of observation when rainfall is larger than 5mm/day. In general, the PDF of WRF-simulated rainfall is more similar to that of observation than that of ERA5.

The spatial distribution of biases in average summer temperature simulated by WRF relative to observation is presented in Fig. 6a. The results show that temperature simulated by WRF tends to be lower than observation mainly in the upstream of the

YRB. In most places of the YRB, the biases of temperature simulated by WRF range from -10% to 10%. The qq-plot of observed basin-averaged temperature versus simulated basin-averaged temperature in Fig. 6b shows that the distribution of basin-averaged temperature simulated by WRF is linearly correlated with that of observation. Figure 6c presents the basin-averaged summer temperature processes of observation, ERA5 and WRF-simulation. The summer temperature process simulated by WRF always falls within the spread between observation and ERA5 data from June to August. However, systematic biases are observed for temperature. Similar results have also been found in other studies. For example, Zhang et al. (2017) found that there was a cold temperature bias in the eastern China when simulated by the WRF model, and the bias was up to 5℃ in some regions. Yan et al. (2021) also showed that the WRF model produced large cold bias over whole China with the exception of the north-western Xinjiang.

To better validate the model performance, biases of the LHF, SHF and PBLH simulated by WRF are further analysed using ERA5 as the benchmark since no such observations are available in the study. Figure S1 presents the spatial distributions of biases in the LHF, SHF and PBLH simulated by WRF relative to ERA5. The LHF simulated by WRF is lower than ERA5 for most places of the YRB. The most significant biases are mainly observed in the upstream of the YRB (Fig. S1a). For most places of the YRB, biases of LHF range from -40% to 0. For the SHF (Fig. S1b), the negative biases are mainly observed in the upstream of the YRB, while the positive biases are mainly observed in the east of the YRB. For the PBLH (Fig. S1c), the positive biases are mainly observed in the upstream of the YRB, ranging from 20% to 100%. The negative biases are mainly observed in the midstream and downstream of the YRB, ranging from -80% to -20%. For most places of the YRB, biases of PBLH range from -20% to 20%. Although the absolute percent biases of these three variables between simulated data and ERA5 data are large than 20% in some places of the YRB, it does not mean that the model is not properly configured, as biases exist between observed data and ERA5 data and sometimes the biases are large (Gleixner et al., 2020; Tarek et al., 2020). For example, Al-Falahi et al. (2020) showed that the percent bias of average annual precipitation of ERA5 and ground stations was -88.97% over the Al Mahwit governorate in Yemen. Moreover, the simulated data are closer to the observation than the ERA5 data for the rainfall and temperature.

### 4.2 The impacts of LULCC between the 2010 and 1990 on the summer rainfall

Figure 7 shows the differences in the average summer rainfall and extreme summer daily rainfall in YRB between the 2010 and 1990 scenarios. According to the results, the average summer rainfall differences vary from -200 mm to 200 mm over the YRB (Fig. 7a). In most places of the YRB, the average summer rainfall decreases for the 2010 scenario compared with the 1990 scenario. The increases in average summer rainfall are mainly observed in the upstream and midstream. Compared with the average summer rainfall, the changes in 90th percentile summer daily rainfall between the 1990 and 2010 scenarios show a similar spatial distribution (Fig. 7b), while the 99th percentile show a slightly different spatial distribution between two

scenarios (Fig. 7c). For example, the 90th percentile summer daily rainfall increases up to 10 mm, mainly observed in the upstream and midstream, while the 99th percentile summer daily rainfall increases up to 50 mm, mainly observed in the midstream and downstream. Besides, the changes in 99th percentile summer daily rainfall are more significant than those in 90th percentile summer daily rainfall.

Furthermore, changes in rainfall between the 2010 and 1990 scenarios are analysed based on two types of area average: one is the area average based on all grids of the whole YRB (ALL-YRB), and the other is based on only the grids where the population density is greater than 100 per square kilometres (PDG-YRB). There are 3625 grids of the PDG-YRB out of 7935 grids of the whole YRB. The spatial distributions of grids of the PDG-YRB are displayed in Fig. 8. Figure S2 presents the changes in average summer rainfall and extreme summer daily rainfall between the 2010 and 1990 scenarios for ALL-YRB and PDG-YRB. Similarly, Figure S3 presents the changes in maximum 1-, 3- and 5-day summer rainfall between the 2010 and 1990 scenarios. The results show that for most of the years, the rainfall statistics decrease from the 2010 scenario to the 1990 scenario. Moreover, the variation ranges of all statistics are always larger for PDG-YRB than for ALL-YRB.

To further understand the responses and sensitivities of summer rainfall to the impacts of LULCC, PDFs of average summer rainfall are shown in Fig. S4a and S4b for 1990 and 2010 scenarios, respectively. In general, the PDF of the 2010 scenario is higher than that of the 1990 scenario for both ALL-YRB and PDG-YRB. Moreover, the PDF of the same scenario (1990 or 2010 scenario) is higher for PDG-YRB than for ALL-YRB. Figure S4c and S4d present the relative changes in multiyear-averaged monthly rainfall during the summer period between the 2010 and 1990 scenarios for both ALL-YRB and PDG-YRB. It can be found that the summer rainfall for the 2010 scenario decreases compared with the 1990 scenario as the relative changes from June to August are all negative.

### 4.3 The impacts of reforestation on the regional climate in the YRB

### 4.3.1 Changes in the summer rainfall

Figure 9a and 9b show the spatial changes in the average summer rainfall between the 20% scenario and 2010 scenario, and between the 50% scenario and 2010 scenario, respectively. From the results, we can see that the average summer rainfall shows a large spatial heterogeneity over the study area. For the 20% scenario, the increases of average summer rainfall (up to 200 mm) are observed in most places of the YRB, while the decreases (up to -100 mm) are mainly observed in the upstream region. For the 50% scenario, the most significant increase in average summer rainfall is observed in the upstream of the YRB, while the most significant decrease is observed in the midstream region. When comparing the changes in average summer rainfall between the 20% and 50% scenarios, areas with an increase in average summer rainfall are more expansive for the 20% scenario than for the 50% scenario.

The spatial distributions of the changes in 90th percentile summer daily rainfall (Fig. 9c and 9d) are similar to those in average summer rainfall for both 20% and 50% scenario. The changes in 90th percentile summer daily rainfall range from -10 mm to 10 mm. Figure 9e and 9f show the changes in the 99th percentile summer daily rainfall between the 20% scenario and 2010 scenario, and between the 50% scenario and 2010 scenario, respectively. For the 20% scenario, the 99th percentile summer daily rainfall increases in most places of the YRB, while the decreases are mainly observed in the midstream. For the 50% scenario, the most significant increase in the 99th percentile summer daily rainfall (up to 50 mm) is mainly observed in the upstream of the YRB, while the most significant decrease (up to -50 mm) is mainly observed in the midstream. Besides, the decrease of the 99th percentile summer daily rainfall for the 50% scenario (up to -50 mm) is more significant than that for the 20% scenario (up to -40 mm). The above results indicate that the average summer rainfall and extreme summer daily rainfall are sensitive to reforestation (conversion from cropland to forest).

### 4.3.2 Area average changes in rainfall

Figure 10 presents the changes in average summer rainfall and extreme summer daily rainfall between the two hypothetical reforestation scenarios and the 2010 scenario for ALL-YRB and PDG-YRB. For most of the years, the average summer rainfall increases for both hypothetical reforestation scenarios comparing with the 2010 scenario. The mean values of changes in extreme summer daily rainfall among ten years also show that all the extreme indices increase for both hypothetical reforestation scenarios comparing with the 2010 scenario. The median values of the changes in all indices are more significant for the 20% scenario than for the 50% scenario for both ALL-YRB and PDG-YRB. Furthermore, the variation ranges of average and extreme summer daily rainfall are always larger for PDG-YRB than for ALL-YRB. In other words, the impacts of reforestation are more significant in the populated area.

Figure 11 presents the changes in maximum 1-, 3- and 5-day summer rainfall between the two hypothetical reforestation scenarios and the 2010 scenario. For the maximum 1-day rainfall, the median values of the 20% scenarios are positive, while those of the 50% scenario are negative. The maximum 3- and 5-day summer rainfall increase for most of the years for both hypothetical reforestation scenarios comparing with the 2010 scenario. Moreover, the median values of the changes in all indices are larger for the 20% scenario than for the 50% scenario for both ALL-YRB and PDG-YRB. Besides, the impacts of reforestation are also more significant in the populated area than over the whole basin.

To indicate more clearly the responses and sensitivities of summer rainfall to the impacts of reforestation, the PDFs of average summer rainfall for the three scenarios (i.e., 2010, 20% and 50% scenarios) are shown in Fig. 12a and 12b. Figure 12a presents the PDFs of average summer rainfall for the three scenarios for ALL-YRB. The PDFs of rainfall for the three scenarios are pretty similar averaged for ALL-YRB, except for the light rainfall of 2 ~ 4 mm/day, which is more for the 2010 scenario than for the 20% scenario and 50% scenario. Figure 12b presents the PDFs of average summer rainfall for three scenarios for PDG-

YRB. The PDF of rainfall for the 2010 scenario is higher than that for the 20% and 50% scenarios when rainfall is smaller than 4mm/day. The PDF for the 20% scenario is higher than that for the 50% scenario when rainfall is about 2 – 4 mm/day.

Moreover, the PDF for the 20% scenario is higher than that for the 2010 scenario when rainfall is around 5.5-7.5 mm/day. Figure 12c and 12d present the relative changes in multiyear-averaged monthly rainfall during the summer period between the two hypothetical reforestation scenarios and the 2010 scenario for both ALL-YRB and PDG-YRB. For rainfall for both ALL-YRB and PDG-YRB, all the relative changes for the 20% and 50% scenarios are positive. The relative changes in rainfall for the 20% scenario are more significant than those for the 50% scenario, except in June for ALL-YRB. Furthermore, the relative

changes for PDG-YRB are more significant than those for ALL-YRB for all months in summer. The results indicate that (1) the reforestation, no matter for the 20% or 50% scenarios, increases summer rainfall; (2) Under the impact of the reforestation, the 20% scenario results in a more significant increase of summer rainfall than the 50% scenario; (3) The impacts of reforestation on average monthly rainfall during the summer period are more significant for the populated area.

**4.3.3 Changes in the latent heat flux, sensible heat flux, and PBL height**

The changes in the LHF, SHF, and PBLH are investigated after analysing the changes in rainfall under reforestation. Figure 13a and 13b show the changes in LHF between the 20% scenario and 2010 scenario, and between the 50% scenario and 2010 scenario, respectively. The spatial distributions of LHF changes for the 20% and 50% scenarios are similar. For example, the LHF increases in most places of the southeastern YRB and decreases in most places of the upstream and midstream for both scenarios. The most significant increases of LHF (up to 20 W/m$^2$) are also mainly observed in the southeastern YRB for both

scenarios. From the results of the significance test in Fig. 13a and 13b, it can be found that the increases of LHF were more significant than decreases after reforestation. The changes in SHF have similar spatial distribution for both 20% and 50% scenarios (Fig. 13c and Fig. 13d). The SHF decreases in many places of the YRB, while the increases of SHF are mainly observed in the north YRB. The largest SHF decreases up to -15 W/m$^2$ are mainly seen in the southeastern YRB. However, areas with increased SHF are more for the 50% scenario than for the 20% scenario. Moreover, through a quantitative

investigation in changes of LHF and SHF over the whole basin, it can be found that the basin-averaged summer daily LHF increases by 0.26 and 0.61 W/m$^2$ for the 20% and 50% scenarios, respectively, while the basin-averaged summer daily SHF decreases by 0.54 and increases by 0.54 W/m$^2$ for the 20% and 50% scenarios, respectively. Figure 13e and 13f show the changes in PBLH between the 20% scenario and 2010 scenario, and between the 50% scenario and 2010 scenario, respectively. The spatial distributions of PBLH change are similar for both 20% and 50% scenarios. Nevertheless, areas with PBLH

increases by more than 30 m or decreases by more than -30 m is more for the 50% scenario than for the 20% scenario. The changes in SHF caused by reforestation can alter the thermodynamic variable PBLH. From Fig. 13, it can be seen that the changes in SHF and PBLH have a similar spatial pattern. When the SHF increases, it leads to the increases of PBLH, which

will increase the possibility of cloud formation and finally enhance the intensity and frequency of extreme rainfall (Shem and Shepherd, 2009).

### 4.3.4 Changes in the 2m air temperature, relative humidity and water vapor mixing ratio

The changes in the 2m air temperature, 2m relative humidity and 2m water vapor mixing ratio under reforestation are also analysed. Figures 14a and 14b present the average summer temperature changes between the 20% scenario and 2010 scenario, and between the 50% scenario and 2010 scenario, respectively. For the 20% scenario, the average summer temperature decreases in most places of the YRB, while the decreases are mainly observed in the central YRB. In only a few small areas in the source region and the eastern part of the YRB, the temperature increases. For the 50% scenario, areas with decreased average summer temperature are reduced compared with that for the 20% scenario. The maximum drop in the summer temperature is -0.8 °C for the 20% scenario and -0.6 °C for the 50% scenario. Meanwhile, there are significant differences between the two scenarios in some regions. For example, in the north of the central YRB, the average summer temperature increases in the 50% scenario while decreases in the 20% scenario. Moreover, changes in surface skin temperature are also analysed, and the results are almost the same as changes in 2m air temperature (Fig. S5).

Figure 14c and 14d present the changes in 2 m relative humidity between the 20% scenario and 2010 scenario, and between the 50% scenario and 2010 scenario, respectively. We can find that the relative humidity changes for these two hypothetical reforestation scenarios have different spatial distributions from the figures. For instance, for the 20% scenario, the relative humidity increases in most places of the YRB with the most significant increases (up to 6%) in the central YRB. When comparing Fig. 14c and Fig. 14d, it can be seen that more areas with increased relative humidity can be found in the 20% scenario than in the 50% scenario. Furthermore, the relative humidity decreases in the north of the central YRB in the 50% scenario, which is not observed in the 20% scenario. Figures 14e and 14f present the changes in the 2m water vapor mixing ratio between the 20% scenario and 2010 scenario and between the 50% scenario and 2010 scenario, respectively. The results show that the water vapor mixing increased at 2m, especially for the 20% scenario. For the 50% scenario, areas with the significant water vapor mixing ratio increased were more than areas with significant water vapor mixing ratio decreased.

From the changes in the 2m air temperature and 2m relative humidity under reforestation, it can be seen that the 2m relative humidity decreases where the 2m air temperature increases. Besides, the water vapor mixing ratio in the atmosphere increases, which finally provides conditions for the increases of summer rainfall amount and extremes.

## 5 Discussions

Comparing the WRF modelling results with observation data, the summer rainfall from the WRF model tends to have positive biases in the north-western YRB while negative biases in the south-eastern YRB. The explanations are that on the one hand,

the upstream of the YRB is a mountainous region, where has only few rainfall stations and many of the rainfall stations located

in the valley, which may result in an underestimation of the rainfall. On the other hand, the resolution of topography used in

the model of 15 km probably impacts the performance of rainfall. Previous studies found that the drag forces of the mesoscale

(3-10 km) and microscale (<3 km) orography would prevent the moisture flux from being taken to the high-altitude complex

terrain region (Wang et al., 2020). However, in our studies, the horizontal resolution of the inner domain is 15 km, which

cannot take the mesoscale and microscale orography into account. Thus, the drag forces of the terrain are diminished, and

more moisture is taken from the low-altitude region (i.e., the south-eastern YRB) to the high-altitude region (i.e., the upstream

of the YRB), which finally causes that the simulated rainfall tends to have positive biases in the high elevation area over the

upstream of the YRB, and negative biases in the low elevation area over the south-eastern YRB. We acknowledge that there

are uncertainties from the bias of WRF modelling in the study. However, the WRF model can still be used to investigate the

impacts of LULCC and reforestation on summer rainfall. The only difference in the initial conditions used to force the WRF

model for the four scenarios is the land cover. Thus, the changes in summer rainfall between different scenarios can be

considered as the results of LULCC.

The changes in average summer rainfall show a sizeable spatial heterogeneity between the 1990 and 2010 scenarios, while the

99th percentile summer daily rainfall shows significant increases in most places of the midstream and downstream of the YRB.

The land use changes from 1990 to 2010 involve not only the increase of forests but also the change of other land uses. Table

1 shows that from 1990 to 2010, the area of cropland decreases from 29.15% to 28.48% for the whole basin, the area of forest

increases from 42.82% to 43.60%, the area of grassland decreases slightly from 23.50% to 23.13%, the area of water and

wetland increases slightly from 1.65% to 1.79%, the area of urban increases from 0.19% to 0.86%, and the unused land

decreases from 2.69% to 2.14%. Therefore, although the forests increase between 1990 and 2010, the rainfall decreases with

the joint impacts of all other land use changes. Furthermore, the main LULCC in the midstream and downstream of the YRB

between the 1990 and 2010 scenarios is the rapid expansion of the urban area. Therefore, it can be inferred that urbanization

may increase the intensity of 99th percentile summer daily rainfall at a local scale. Similar results can be found in other studies.

For example, Wang et al. (2015) found that extreme rainfall events had a strong positive spatial correlation with the urban

extent. Zhang et al. (2018) also found that urbanization led to an amplification of the total rainfall along with a shift in the

location of the maximum rainfall and further increased the intensity and frequency of extreme flooding events. However, from

the basin-averaged results, both the average summer rainfall and extreme summer rainfall decrease for the 2010 scenario

compared with the 1990 scenario.

The rainfall changes between the two hypothetical reforestation scenarios (20% and 50% scenarios) and the 2010 scenario

show that transforming cropland to forest increases summer rainfall. However, transforming different proportions of cropland

to forest has different impacts on local rainfall. With the transform proportion of cropland to forest increases from 20% to 50%, the summer rainfall increases less conversely. To better explain this result, the changes in the water vapor mixing ratio at 2m (Fig. 14), upward moisture flux at the surface (Fig. S6) and wind at 10m (Fig. S7) are further analysed. It can be found that the number of grids showing increased upward moisture flux in the 50% scenario slightly exceeds that in the 20% scenario. In contrast, the 2m water vapor mixing ratio increases over almost all basin in the 20% scenario while shows significant decreases in the midstream of the basin in the 50% scenarios. From the surface level to the 2m level, the moisture keeps increased in the 20% scenarios while decreases in the 50% scenarios. This suggests that the distribution of moisture may be changed by the horizontal transportation processes. Furthermore, Fig. S7 shows that the 10m wind decreases in most places of the Yangtze River Basin for both scenarios, which is as expected because reforestation increases the surface roughness. However, the 10m wind increases around the reforested areas, accelerating the moisture export from the forest. It is worth noting that areas with an increase in 10m wind are more expansive for the 50% scenario than for the 20% scenario, which means that more moisture is transported from the forest to other places for the 50% scenario. In addition, from the changes in wind direction in Fig. S7, moisture exported from the forest is transported towards the southern regions and finally flow out the Yangtze River Basin. The above analyses further prove that the differences between the 20% and 50% reforestation scenarios are mainly caused by the changes of horizontal wind. Moreover, Yu et al. (2020) found that the vegetation greening reduced rainfall in some region in southern China, which may be caused by the East Asian monsoon, as the East Asian monsoon significantly influenced the summer rainfall patterns in China (Ding et al., 2007). Furthermore, comparing the simulation results from the whole YRB with the results from grids where the population density is greater than 100 per square kilometres, it can be seen that the impacts of LULCC on summer rainfall are more significant for the populated area, which means that reforestation will have significant impacts on human lives.

Although a comprehensive assessment of the impacts of LULCC on summer rainfall amount and extremes was conducted in this study, some issues remained. For example, only one regional climate model (i.e., the WRF model) was used in this study, although it has been widely used in China (Huang et al., 2020; Azmat et al., 2020). Some previous studies indicated that results from a single RCM had a significant uncertainty since RCMs could perform differently in the same region (Davin et al., 2020; Zhang et al., 2017). In this case, it is worthwhile to look at the impacts of LULCC on summer rainfall and extremes based on an ensemble of multiple RCMs in future researches. We also aware that convective parameterizations differ significantly in their treatment of the cloud up draughts and down draughts, mass-flux closure and triggering, often assuming that one is averaging over both cloud up draughts and the subsiding environment. As a result, all these schemes are better at predicting the area-average rainfall (Clark et al., 2016). Additionally, the cumulus parameterizations also introduce uncertainties to the model results (Liu et al., 2016). Besides, regarding the WRF spatial resolution impacts, we used 15 km in the study regarding

a large nested domain focusing on the Yangtze River basin with a total area of ~$1.8 \times 10^6$ km$^2$. The modelling resolution in the study was comparable with other studies that investigate the impacts of land use/cover changes over a large region (e.g., Zhang et al., 2017; Zha et al., 2019; Zhang et al., 2021). However, we acknowledged that higher model resolution of WRF simulation, e.g., convection-permitting scale, may better represent rainfall processes and land surface (Knist et al., 2020; Kurkute et al., 2020).

Moreover, there were 32 eta levels of the model, and the top was at 50 hPa. We acknowledged that we did not test whether there were enough layers near the bottom to trust the surface values. However, many relevant studies used similar or less vertical levels to study the changes of these surface variables (e.g., Hu et al., 2015; Yu et al., 2020). Moreover, Gallus et al. (2009) found that doubling the number of vertical levels from 31 to 62 did not result in a consistent improvement in the rainfall forecasts. The skill might not be improved much by refining the number of levels, although we acknowledge that the finding from Gallus's study may be different if it is in a different study area. On the other hand, adding the number of levels requires more computing resources and running time, which will limit what we can achieve in the study regarding it, since it is already quite heavy to finish around 40 years WRF simulations with such a big nested domain. Furthermore, we didn't use urban scheme in the WRF modelling in the study. However, the urban area was only 0.19 % of the total area in 1990 and increased to be 0.86% in 2010. In this case, the impact of urbanization is ignorable in the study, regarding the increased urban area is only around 0.67% of total area of YRB from 1990 to 2010.

Furthermore, the random sampling method was used to produce the two hypothetical reforestation scenarios in this study. Thus, the grids where the cropland was changed to be forest tended to be distributed evenly among the croplands in the YRB. However, the reforestation process usually happened in specific areas that were relative to local policy. It was challenging to gather the related policies from multiple local governments over such a big basin. It could also be noticed that the crops were mainly located in specific areas such as the Sichuan Basin and the middle and downstream of the YRB. Although we randomly chose the crop grids, the restoration grids concentrated on these specific areas similar to the actual reforestation processes. Despite these, this study could still provide a sight of what would happen to summer rainfall under reforestation.

## 6 Conclusions

In this study, analysis based on the WRF model simulations was used to research the impacts of LULCC and reforestation on summer rainfall amount and extremes in the YRB. Two observed scenarios (1990 and 2010 scenarios) were chosen to compare and investigate the changes in summer rainfall under the impacts of LULCC during the last decades. Besides, two hypothetical reforestation scenarios (20% and 50% scenarios) produced based on the 2010 scenario were used to test the sensitivity of

summer rainfall to reforestation. The changes in summer rainfall between different scenarios were analysed, and the potential mechanisms were discussed. The main conclusions are outlined below:

1. LULCC largely influenced summer rainfall amount and extremes during 1990-2010 in the YRB. The LULCC between the 1990 and 2010 scenarios decreases average summer rainfall. Although the extreme summer daily rainfall increases up to 50 mm in some places of the midstream and downstream, the overall pattern is decrease for the whole basin.

2. Reforestation can affect heat flux, air temperature, relative humidity and PBLH in the YRB, leading to more water vapor mixing in the atmosphere, which provides conditions for the increases of summer rainfall amount and extremes. Moreover, the effects of reforestation are more pronounced in the populated area than over the whole basin.

3. Although reforestation increases summer rainfall both in the total amount and extremes, the differences exist in the scenarios with different reforestation proportions of 20% and 50%. Specifically, with the transform proportion of cropland to forest

increases from 20% to 50%, the summer rainfall increases less conversely. By analysing the changes in water vapor mixing ratio, upward moisture flux and 10m wind, it suggests that this result may be caused by the horizontal transportation processes of moisture.

**Author contributions**

JC, LL and WL designed the overall study. JC and HC collected the data. WL and LL developed the model code and performed
the simulations, with some contributions from QL. WL, JC and LL contributed to the interpretation of results. WL wrote the paper, and JC and LL revised the paper.

**Competing interests**

The authors declare that they have no conflict of interest.

**Acknowledgements**

This work has been partially supported by the Hubei Provincial Natural Science Foundation of China (Grant No. 2020CFA100), the National Natural Science Foundation of China (Grant No. 52079093, 51779176), the Overseas Expertise Introduction Project for Discipline Innovation (111 Project) funded by Ministry of Education and State Administration of Foreign Experts Affairs P.R. China (Grant No. B18037), and the Center for Climate Dynamics (SKD) through the Bjerknes Centre for Climate Research (CHEX and 100878-FTI). We thank Priscilla Mooney (NORCE) for providing her suggestions in the early analysis.
The numerical calculations in this paper have been done on the supercomputing system in the Supercomputing Center of Wuhan University.

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

**Tables**

**Table 1. The percentages of land use classes under four scenarios.**

| Scenarios | Cropland (%) | Forest (%) | Grassland (%) | Water and wetland (%) | Urban (%) | Unused land (%) |
|---|---|---|---|---|---|---|
| 1990 scenario | 29.15 | 42.82 | 23.50 | 1.65 | 0.19 | 2.69 |
| 2010 scenario | 28.48 | 43.60 | 23.13 | 1.79 | 0.86 | 2.14 |
| 20% scenario | 22.80 | 49.28 | 23.13 | 1.79 | 0.86 | 2.14 |
| 50% scenario | 14.58 | 57.50 | 23.13 | 1.79 | 0.86 | 2.14 |

**Figures**

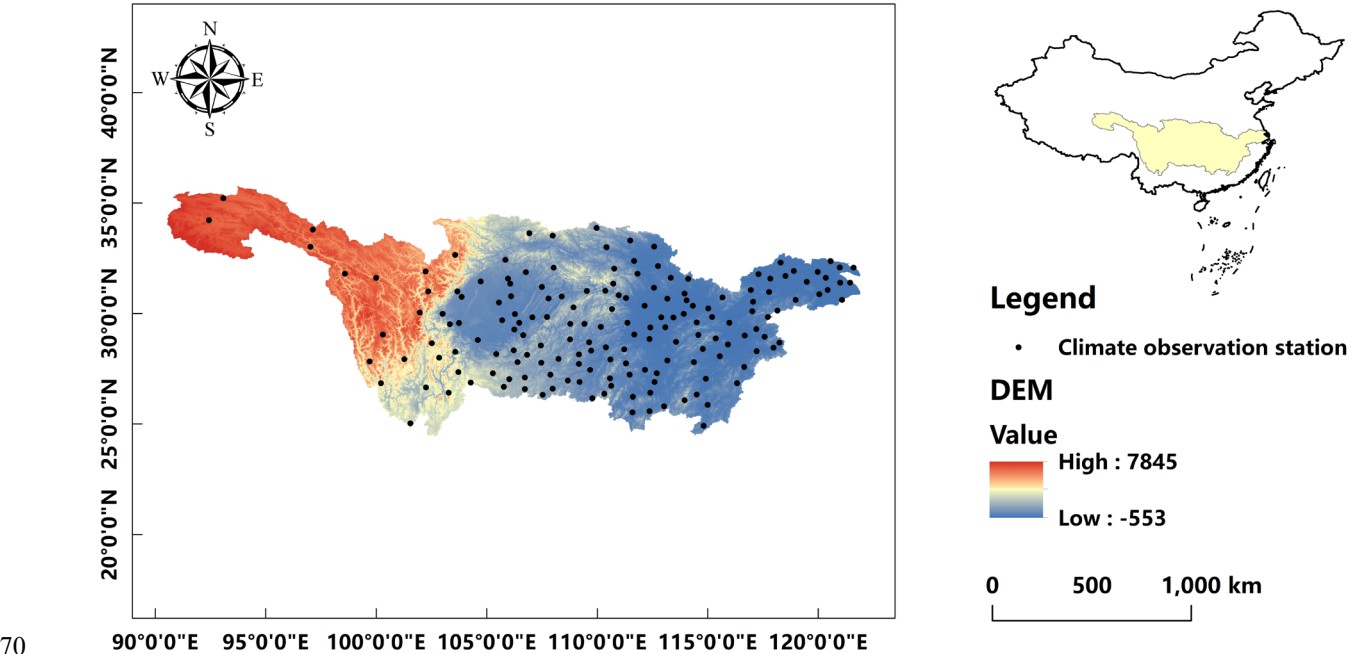

**Figure 1. The location and topography of the Yangtze River basin and the location of climate observation stations.**

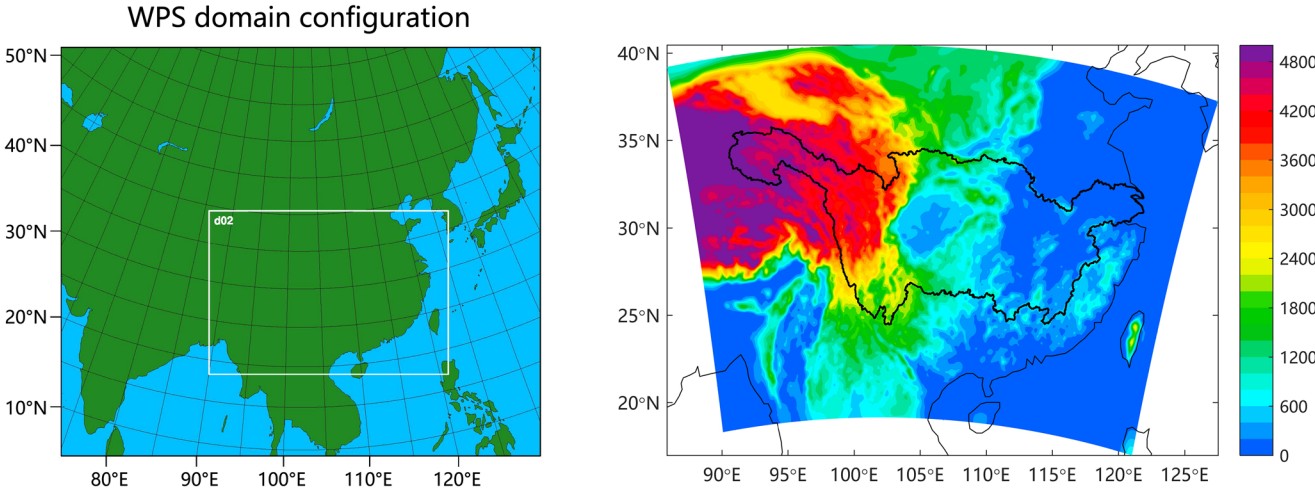

**Figure 2. The WRF model domain and the model topography (units: m).**


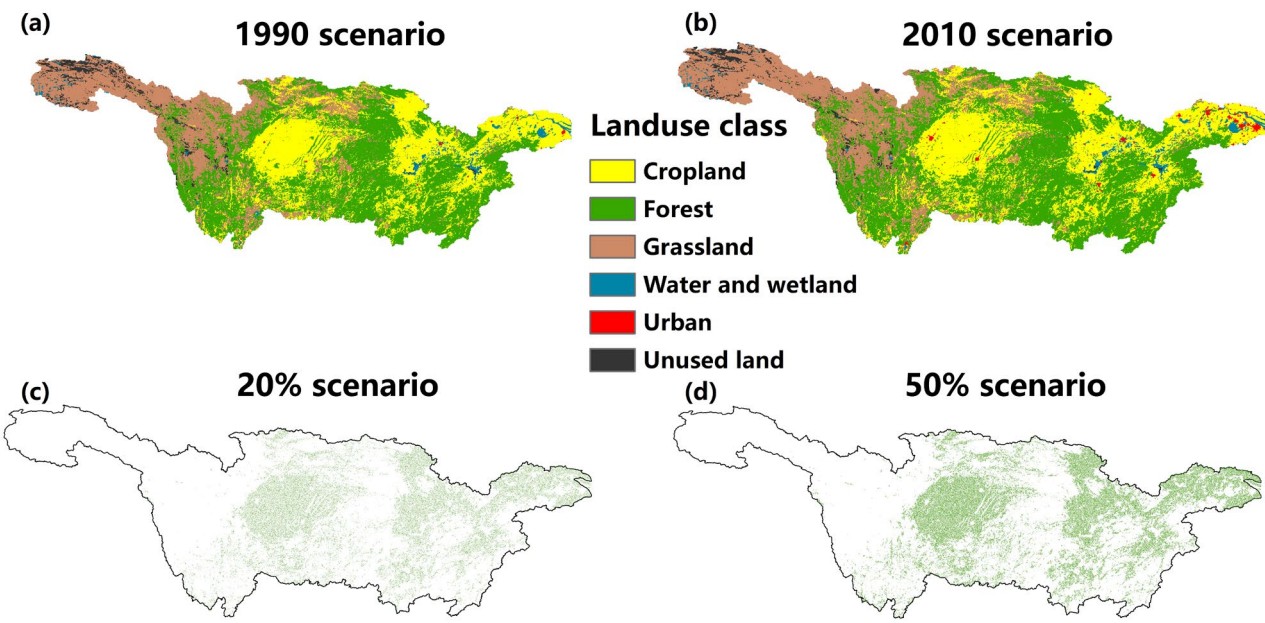

**Figure 3. (a-b) Land use and cover under 1990 and 2010 scenarios; (c-d) Land use and cover changes between the two hypothesis scenarios (20% and 50% scenarios) and 2010 scenario.**

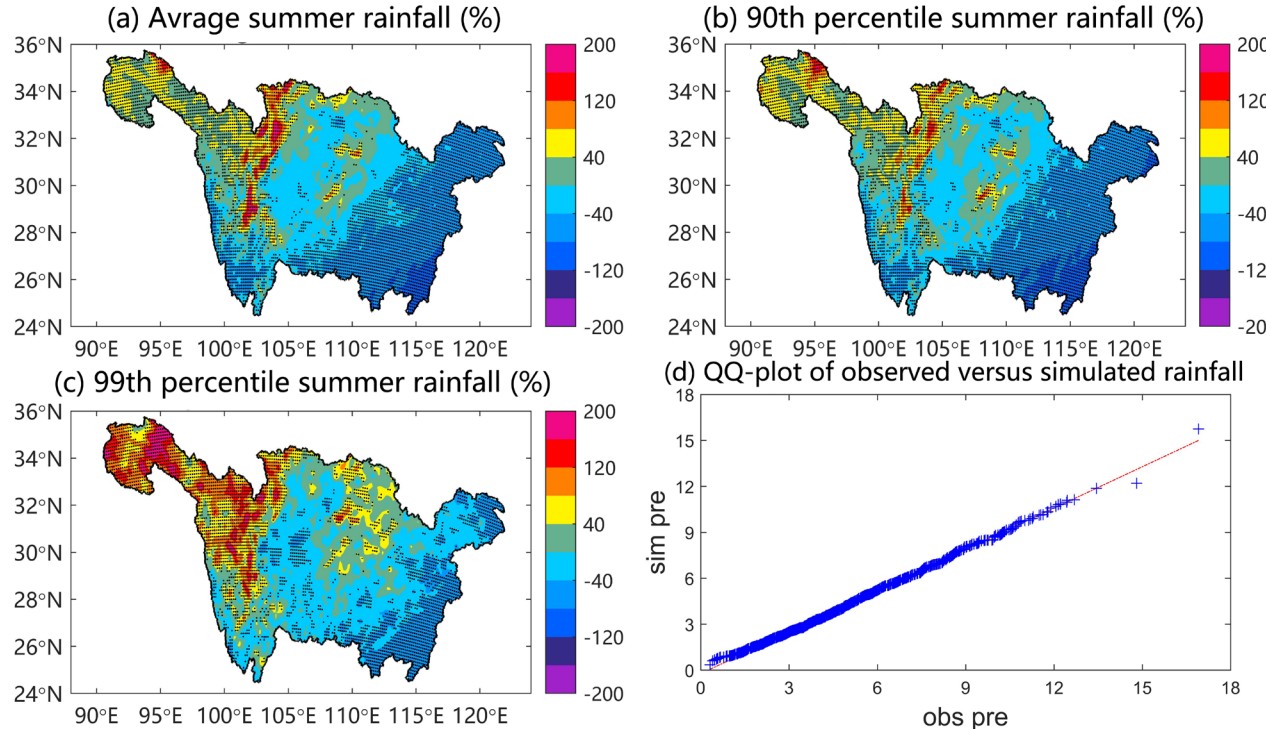


**Figure 4. The bias of (a) average summer rainfall (%), (b) 90th percentile summer rainfall (%) and (c) 99th percentile summer rainfall (%) between the 2010 scenario and observed data, and (d) the qq-plot of observed rainfall versus simulated rainfall. The stippling regions show statistically significance of bias identified by t-test at a 5% significance level.**

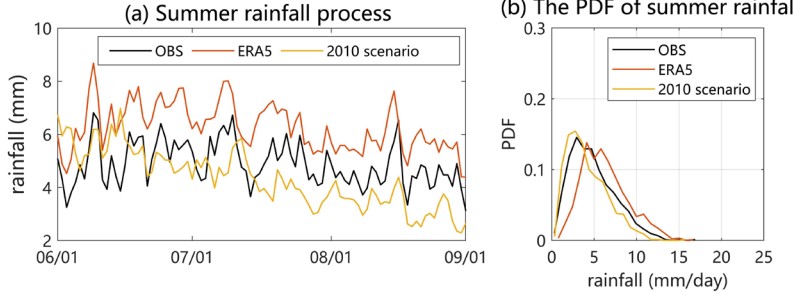


**Figure 5. (a) The basin-averaged summer rainfall processes of observation, ERA5 and 2010 scenario; (b) The probability distribution functions of summer rainfall of observation, ERA5 and 2010 scenario.**

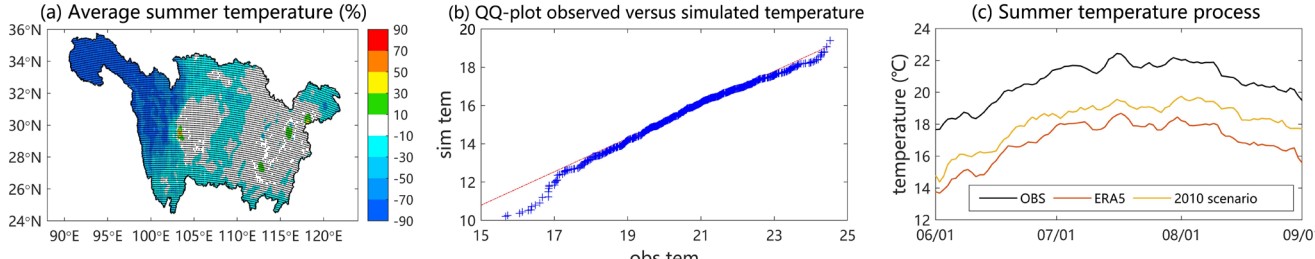

**Figure 6. (a) The biases of average summer temperature (%) between the 2010 scenario and observed data, the stippling regions show statistically significance of bias identified by t-test at a 5% significance level.; (b) The qq-plot of observed temperature versus simulated temperature; (c) The basin-averaged summer temperature processes of observation, ERA5 and 2010 scenario.**

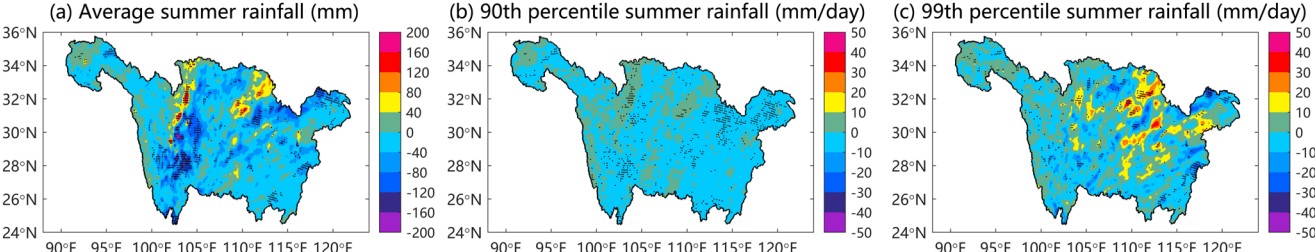

**Figure 7. The changes in (a) average summer rainfall (mm), (b) 90th percentile summer rainfall (mm/day) and (c) 99th percentile summer rainfall (mm/day) between the 2010 scenario and 1990 scenario. The stippling regions show statistically significance of changes identified by t-test at a 5% significance level.**

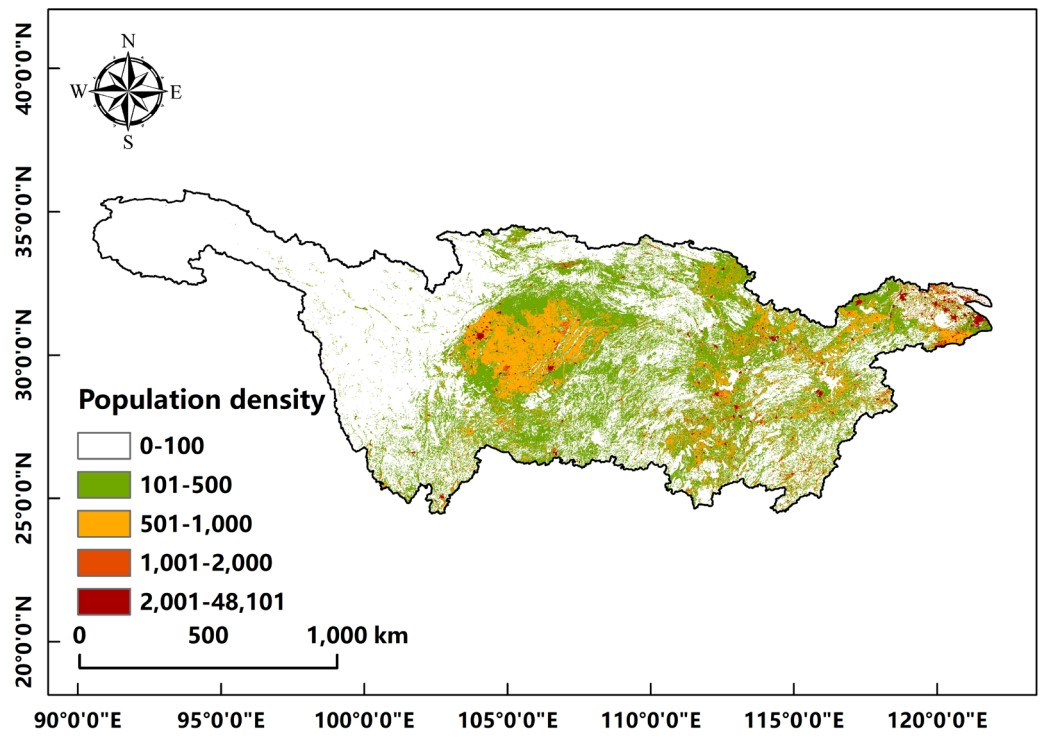

**Figure 8. The spatial distributions of grids where the population density is greater than 100 per square kilometres (PDG-YRB).**

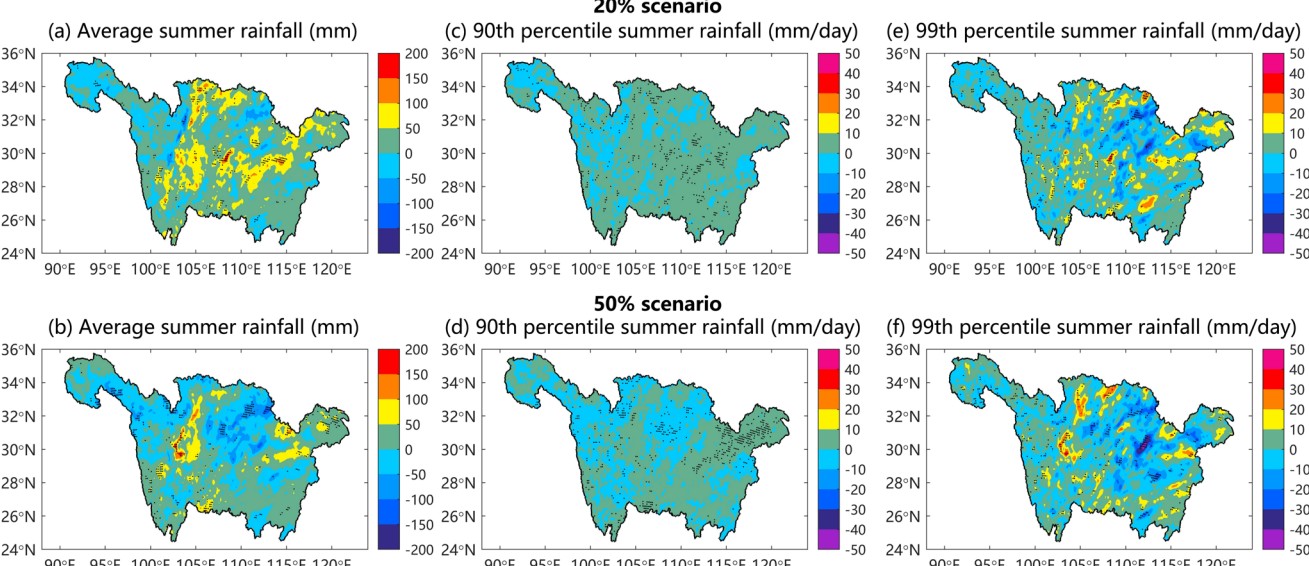

**Figure 9. The changes in (a-b) average summer rainfall (mm), (c-d) 90th percentile summer rainfall (mm/day) and (e-f) 99th percentile summer rainfall (mm/day) between the 20% scenario and 2010 scenario, and between the 50% scenario and 2010 scenario. The stippling regions show statistically significance of changes identified by t-test at a 5% significance level.**

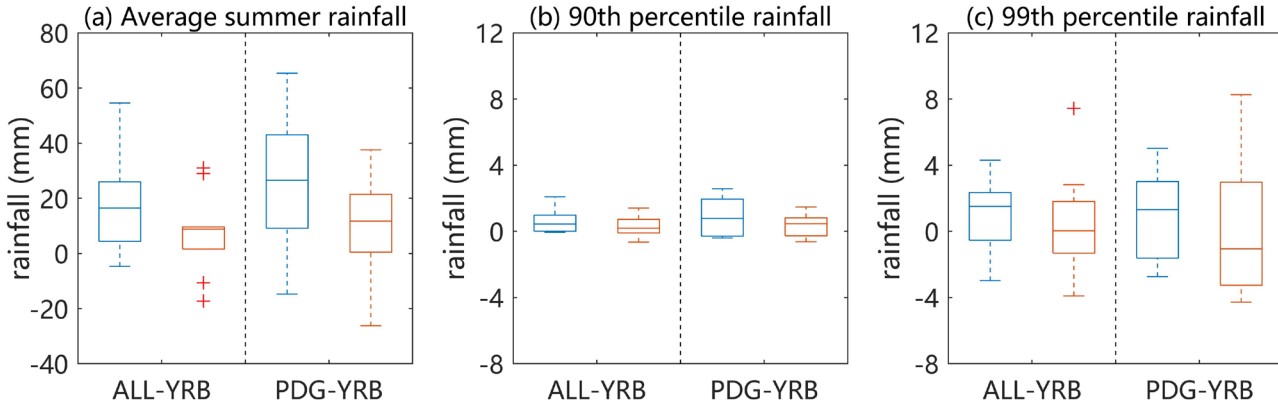

**Figure 10. The changes in (a) average summer rainfall (mm), (b) 90th percentile summer rainfall (mm/day) and (c) 99th percentile summer rainfall (mm/day) between the two hypothesis scenarios (20% and 50% scenarios) and 2010 scenario in ALL-YRB and PDG-YRB area. The blue boxes represent the 20% scenario, while the red boxes represent the 50% scenario.**

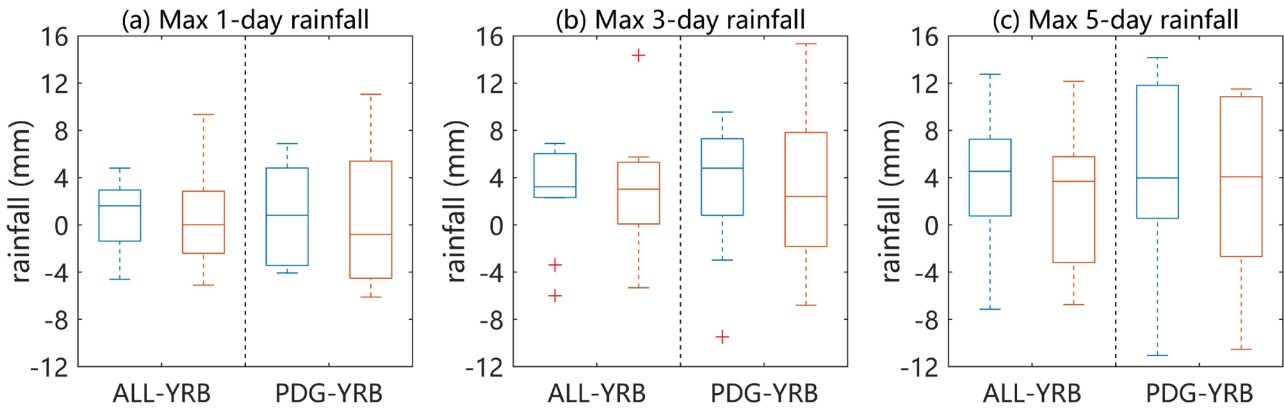

**Figure 11. The changes in maximum 1-, 3-, 5-day rainfall between the two hypothesis scenarios (20% and 50% scenarios) and 2010 scenario in ALL-YRB and PDG-YRB area. The blue boxes represent the 20% scenario, while the red boxes represent the 50% scenario.**

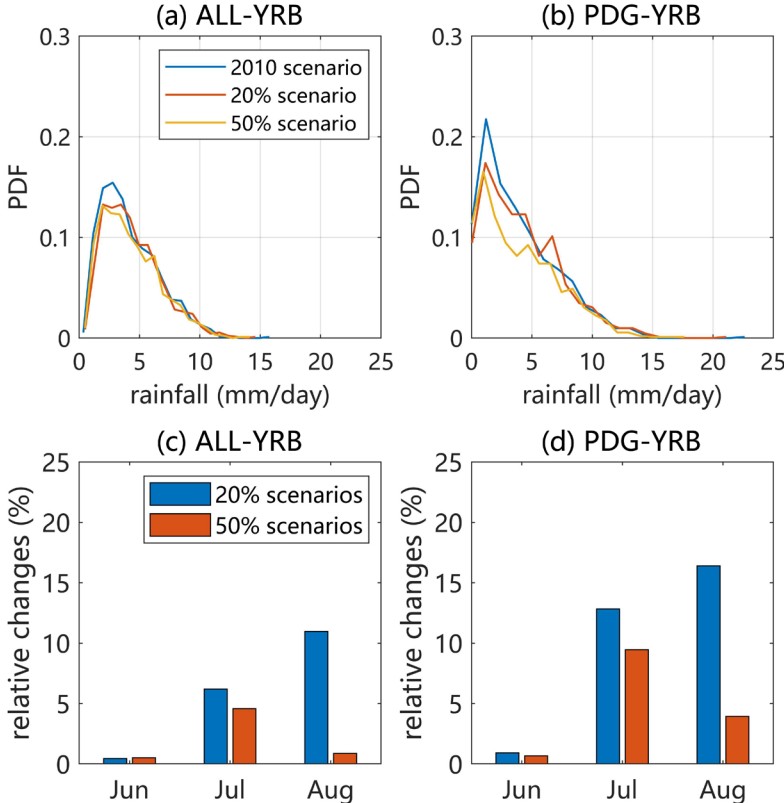

**Figure 12. The probability distribution functions of summer rainfall in 2010, 20% and 50% scenarios in (a) ALL-YRB and (b) PDG-YRB; The changes in multiyear-averaged summer monthly rainfall between the two hypothesis scenarios (20% and 50% scenarios) and 2010 scenario in (c) ALL-YRB and (d) PDG-YRB.**


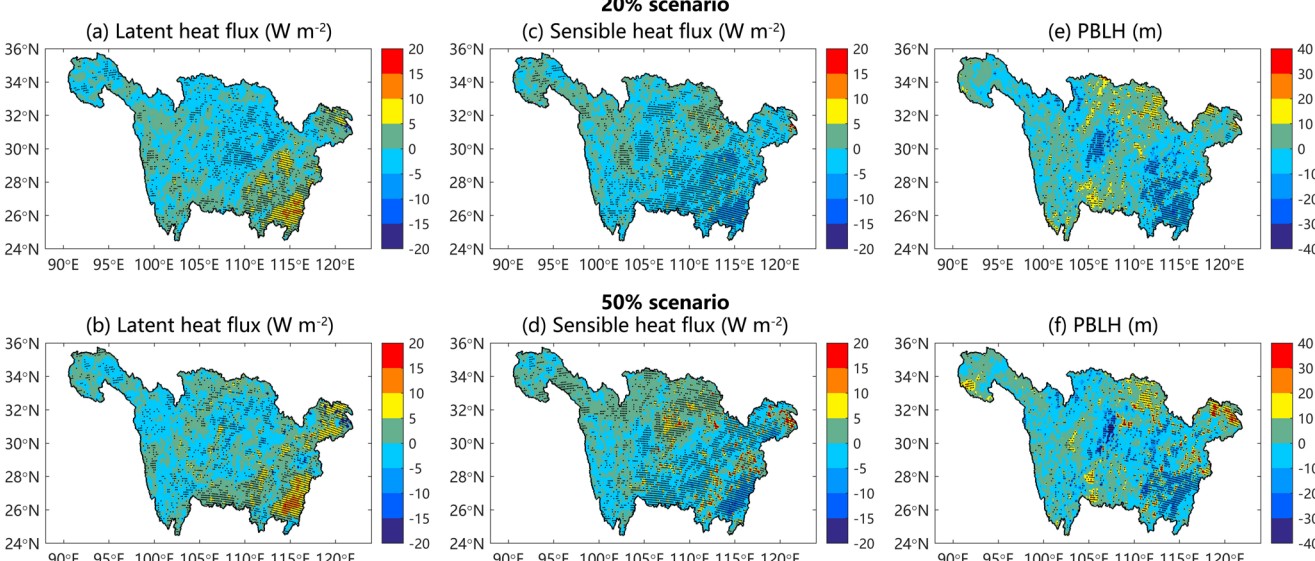

**Figure 13. The changes in (a-b) latent heat flux (LHF, W/m², (c-d) sensible heat flux (SHF, W/m²) and (e-f) PBL height (PBLH, m) between the 20% scenario and 2010 scenario, and between the 50% scenario and 2010 scenario. The stippling regions show statistically significance of changes identified by t-test at a 5% significance level.**

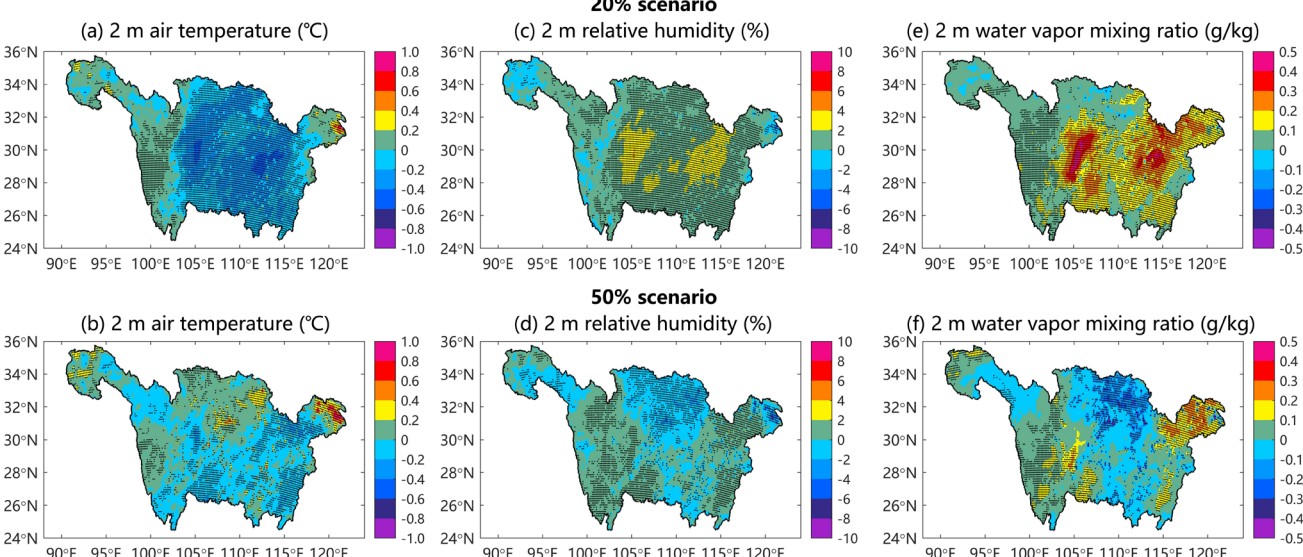

**Figure 14. The changes in (a-b) 2m air temperature (℃), (c-d) 2m relative humidity (%) and (e-f) 2m water vapor mixing ratio (g/kg) between the 20% scenario and 2010 scenario, and between the 50% scenario and 2010 scenario. The stippling regions show statistically significance of changes identified by t-test at a 5% significance level.**