# Peer review of "Impacts of land use/cover change and reforestation on summer rainfall for the Yangtze River Basin"

_Hydrology and Earth System Sciences, 2020_

## Referee Comment (RC1) · Emma Daniels (Referee) · 23 Sep 2020

General comments:

In general, I think the paper has interesting results and could be published. However, the quality of English needs to be improved in some parts (mainly abstract, introduction and methods). Moreover, I miss details in your method such as your definition of summer (i.e. which months are analyzed) and details on the land use maps (e.g. a table with percentages) and how they are included. I think the paper would benefit from analysis of an additional parameter for extreme precipitation, such as rainfall above the 90th percent as with 10 years of data (i.e. 900 data points assuming a summer of 3 months) the 99th percentile alone might be misleading. Furthermore, I miss an expla-

nation why precipitation is thought to increase with further reforestation but decreased between 1990 and 2010 though forest cover increased. Also, I wonder why Shrubland (USGS code 8) and Savanna (USGS code 10) are chosen as a type of forest? Judging from the LANDUSE.TBL these classes are much more similar to Cropland and Pasture than forest, so I wonder if expanding these makes a difference or if you are mainly looking at the effect of the additional Broadleaf forest. I think the figures need work and should become more informative than mainly barplots and spatial difference plots.

Specific comments:

The number and quality of references in the first section of the Introduction is poor. I am sure there is more work done on LUCC changes that is more relevant to your work than done in Burkina Faso and Scandinavia. You can also leave these out as you mention more relevant ones later on.

Adding a table to figure 4 with the percentages of LU classes would be more informative.

How are the 32 vertical levels of the model spread? Are there enough layers near the bottom to trust the surface values you are evaluating such as skin surface temperature and 2-m relative humidity?

In Figure 5c (and others), why not show a qq-plot of model and observed rainfall instead? 50th percentile is not interesting to show and analyze.

It seems urbanizations plays a role in the precipitation decrease between 1990 and 2010. Please consider using an urban scheme in WRF.

Why are the two areas (ALL-YRB) and (CTF-YRB) analyzed separately? Is there a rational in being interested in the converted areas specifically? Is analyzing more populated areas separately more interesting perhaps? As that is where the impact will be felt, not in the new forests.

Line 275-277 please reconsider/rewrite.

---

## Referee Comment (RC2) · Anonymous Referee #2 · 8 Nov 2020

The manuscript by Li et al "Impacts of land use/cover change and reforestation on summer rainfall for the Yangtze River Basin" used the WRF model to investigate how land cover changes and reforestation affect summer rainfall. The research topic is important given the massive ecological projects in China and its climate impact is worth studying. The manuscript is generally well-written, but I also have major comments for the authors.

1. For WRF model simulation, how land cover changes were implemented in the model needs more detailed explanations as different land surface models have different representations of land cover. It is still unclear what surface conditions/variables had been modified for the Noah-MP model to correctly reflect the intended land cover changes. I also have questions about the rationality of the randomly chosen crops for the two

restoration scenarios. 2. When comparing simulation results between different experiments, the authors need to conduct statistical significance tests to determine whether the signal is robust while excluding any noise and random changes which may lead to misinterpretation. 3. I hope the authors could provide more mechanistic explanations of the results. For example, why did the 20% reforestation result in more precipitation changes than the 50% reforestation scenario?

Specific comments: L9: There is another terminology "Grain for Green" frequently used in the literature for "Returning Farmland to Forest Program". Which one is better acknowledged? L130-140: What kinds of WRF experiments have been conducted to compare different schemes/parameterizations, what domain and simulation length was used for the comparison experiments?

L145-149: It is better to also report the quantities of land cover changes between 1990 and 2010.

L148: How did the random changes from cropland to forest being incorporated in the model surface land condition at 15 km resolution? I am not sure whether this choice is necessary. What land variables had been modified to represent the land cover change in WRF model and what are their changes? What types of forest were used in the reforestation experiment? How many grid boxes experienced land cover change?

L170: What about statistical significance levels of these precipitation changes? This needs to be reported for this and other figures as well.

L224: Why did 20% and 50% reforestation grids at the model resolution are different?

L241: For this section, the overall decreases in both LHF and SHF after reforestation were unexpected to me. Not sure if these changes are robust enough. Typical, ET would increase after reforestation, as described in the introduction, so how to explain this result?

L259: What about the changes in near-surface air temperature? For example, 2m air

temperature.

L276-277: Any evidence to support this argument, given the latent heat flux decreased?

L296-297: How many urban grids had changed between 1990 and 2010? Whether urban expansion will affect the entire Yangtze river basin?

L332: Is there actual data to support the increased water vapor mixing?

L335: Why is the precipitation response larger in 20% than in the 50% scenario? There is no related explanation or discussion.

―――――――――――――――――――――――

---

## Referee Comment (RC3) · Anonymous Referee #3 · 28 Nov 2020

The authors of the manuscript "Impacts of land use/cover change and reforestation on summer rainfall for the Yangtze River Basin" present work that show the effects of land use and land cover change on regional climate processes including summer rainfall. The manuscript shows the importance of better understanding these effects and has some interesting discussion points. These types of studies are difficult to do and this is a great start. However, In my opinion, the points outlined in this review need to be addressed for this work to have scientific merit.

General:

1. The methods used to change land cover need to be discussed further as other reviewers have mentioned. The land surface model (Noah-MP) is complex and offers many options to better represent land surface processes. The land surface model is

only mentioned once in the text. Noah-MP contains too many options that need to be carefully chosen for this to be glossed over. Additionally, Noah-MP uses only the dominant land use category when calculating surface fluxes, so at 15km an increase in forest will not matter if it doesn't become the dominant category. This may help explain the inconsistent results between the 20% and 50% reforestation but without more information, it's hard to say.

2. The limitations of using a convective parameterization when investigating rainfall extremes needs to be discussed. In a region with large vertical relief, the choice to use a course resolution for this study should be justified. Convection permitting scales (<4km) not only allow for better representation of precipitation processes, but also better land surface representation (including topography).

3. The model validation is insufficient. Look to Liu et al., 2017, for an example of full model validation. To be specific, I would like to see the figures reworked to show the spatial patterns of rainfall on a seasonal and annual basis in the observations and in the control simulations. Furthermore, the figures should include a representation of percent change in rainfall. A bias of 600mm of rainfall during the summer months is a lot if the average summer rainfall is only 1000mm. This information isn't shown so it's hard to know if the bias is significant. Statistical testing should also be included where appropriate. Additionally, validation of other climatic components that contribute to rainfall (such as the vertical structure of the atmosphere, PBLH, CAPE, CIN) would aid this study. Validation of surface fluxes would also help build a better picture of how well the model can represent this region. There are several eddy-covariance towers in the eastern part of the domain and a comparison of sensible and latent heat flux to those towers would be interesting. Any change that is presented should have an accompanying discussion of validation for that component. Showing Figure 10 but compared to observations would be necessary to see if WRF can capture extreme rainfall.

4. The taylor diagrams are honestly pretty confusing, I would remove them and provide

a table of biases instead. The correlation coefficients are rather low for temperature (the easiest for the model to accurately capture) and lower for rainfall when compared to observations. This leads me to believe that the model isn't configured properly for this region. If the above issues were tackled, then this opinion might change. One way to show that the model is well validated is to show that the temperature and rainfall falls within the spread of observations. Comparison to not only the station data but to an independent gridded dataset (such as ERA5, CRU, etc.) would strengthen this point.

5.The percentiles of rainfall need to be defined better. What does 99th percentile mean in this case? Is it the 99th percentile of rainfall events over the 11 years? Without sub-daily rainfall, I'm not sure that this qualifies as extreme per se. A common extreme rainfall metric is the 99th percentile of daily maximum rainfall (requires sub daily rainfall to properly calculate). In my country, the storms that produce flash flooding often last only a few hours, vs a monsoon type rain that produces flooding from many, many hours of low intensity rainfall. More discussion of rainfall in this region would put this information into context. I would remove the figures that show changes to median rainfall and instead discuss some other metric of interest.

6. All the figures showing change between simulations need to have statistical testing. The figures all look very noisy and some of the changes to precipitation could be because the storms moved, not because more rain fell.

7. Instead of bar graphs, boxplots or violin plots should be shown. This will capture the distribution of the change.

Minor specific points:

The convention I have seen for abbreviating land use and land cover change is LULCC not LUCC.

There are some English language errors in the text, but these don't bother me that much and have been covered by other reviewers.

---

## Author Comment (AC1) · 19 Jan 2021

**Reply to Referee comment 1**

Dear Editors and Reviewers:

5 We would like to thank the editor and all reviewers for their valuable suggestions and comments on the manuscript. These comments have not only improved the quality of the current manuscript, but also are beneficial to our future research in general. All point-by-point responses are presented as follows and we will carefully revise the manuscript based on these comments. For clarity, all comments are given in the original version, while responses are marked in blue.

10 **Emma Daniels (Referee)**

**General comments:**

In general, I think the paper has interesting results and could be published. However, the quality of English needs to be improved in some parts (mainly abstract, introduction and methods). Moreover, I miss details in your method such as your definition of summer (i.e. which months are analyzed) and details on the land use maps (e.g. a table with percentages) and how

15 they are included. I think the paper would benefit from analysis of an additional parameter for extreme precipitation, such as rainfall above the 90th percent as with 10 years of data (i.e. 900 data points assuming a summer of 3 months) the 99th percentile alone might be misleading. Furthermore, I miss an explanation why precipitation is thought to increase with further reforestation but decreased between 1990 and 2010 though forest cover increased. Also, I wonder why Shrubland (USGS code 8) and Savanna (USGS code 10) are chosen as a type of forest? Judging from the LANDUSE.TBL these classes are much

20 more similar to Cropland and Pasture than forest, so I wonder if expanding these makes a difference or if you are mainly looking at the effect of the additional Broadleaf forest. I think the figures need work and should become more informative than mainly barplots and spatial difference plots.

Thanks for the comments. We are sorry for the grammar problems in the manuscript. The manuscript will be proofread by a

25 native English speaker. We have added the definition of summer in the introduction, and the summer defined in this manuscript is from June to August. Table 1 will be added in the revised manuscript to explain the percentages of different land use types in the whole basin. Moreover, the land use changes were included in the WRF model by modifying the geographical static data used in the model which further changed the simulation of subprocesses such as the vegetation phenology, canopy stomatal resistance, runoff and groundwater in the land surface model Noah-MP (Li et al. 2018). Many parameters were used in Noah-

30 MP to describe the characteristics of different land use types, such as albedo, HVT (Top of canopy), LAI (Monthly leaf area index), and VCMX25 (Maximum rate of carboxylation at 25 °C). When the land use changed, these parameters changed accordingly which finally led to the changes in substance and energy exchanges between atmosphere and land surface. The

geographical static data we used in the WPS is *landuse_30s_with_lakes* which was download from the WRF website. The land use data of 1990 and 2010 scenarios were derived from the Landsat thematic mapper (TM) digital images. And then, in the

35    YRB, we replaced the land use data from the *landuse_30s_with_lakes* with the land use data from the Landsat thematic mapper (TM) digital images. Finally, we randomly changed 20% and 50% of the croplands to be forests using the 2010 scenario as a baseline to produce 20% and 50% reforestation scenarios.

Given the comments from other reviewers, the 99.95th percent summer rainfall has been chosen to further analyze the extreme rainfall. Furthermore, the land use changes from 1990 to 2010 were not only involved the increase of forests, but also the

40    change of other land uses, such as urbanization and grassland degradation. Therefore, although the forests increased between 1990 and 2010, the precipitation decreased with the joint impacts of all other land use changes.

Moreover, the land-use categories of the original land use data are defined by Liu et al. (2002, 2005), which are commonly used in China. In this category, there are four kinds of forests which are Forest (Liu code 21), Shrub (Liu code 22), Sparse woodland (Liu code 23) and Cut over land (Liu code 24). When converting the land use type from Liu categories to USGS

45    categories, the four forest categories of Liu were converted to Deciduous broadleaf forest (USGS code 11), Shrubland (USGS code 8), Savanna (USGS code 10) and Savanna (USGS code 10), respectively. That was why Shrubland (USGS code 8) and Savanna (USGS code 10) were chosen as a type of forest. All above information and more clarifications will be added in the method section of the revised manuscript. The figures in the revised manuscript will be improved, and we will also add more informative figures such as qq-plot and significance test in the revised manuscript. We display the revised Fig. 5 as follows;

50    other revised figures will be included in the revised manuscript.

[Figure]

**Figure 5. The bias of (a) average summer rainfall (mm), (b) 99th percentile summer rainfall (mm/day) and (c) 50th percentile summer rainfall (mm/day) between the observed data and 2010 scenario, and (d) the qq-plot of observed rainfall versus simulated rainfall. The stippling regions show statistically significance of bias identified by t-test at a 5% significance level.**

**Table 1. The percentages of land use and cover under four scenarios.**

| Scenarios | Cropland | Forest | Grassland | Water and wetland | Urban | Unused land |
|---|---|---|---|---|---|---|
| 1990 scenario | 29.15 | 42.82 | 23.50 | 1.65 | 0.19 | 2.69 |
| 2010 scenario | 28.48 | 43.60 | 23.13 | 1.79 | 0.86 | 2.14 |
| 20% scenario | 22.80 | 49.28 | 23.13 | 1.79 | 0.86 | 2.14 |
| 50% scenario | 14.58 | 57.50 | 23.13 | 1.79 | 0.86 | 2.14 |

Reference:

Li, J., Chen, F., Zhang, G., Barlage, M., Gan, Y., Xin, Y., and Wang, C.: Impacts of Land Cover and Soil Texture Uncertainty on Land Model Simulations Over the Central Tibetan Plateau, Journal of Advances in Modeling Earth Systems, 10, 2121-2146, https://doi.org/10.1029/2018ms001377, 2018.

Liu, J., Liu, M., Deng, X., Zhuang, D., Zhang, Z., and Luo, D.: The land use and land cover change database and its relative studies in China, Journal of Geographical Sciences, 12, 275-282, https://doi.org/10.1007/BF02837545, 2002.

Liu, J., Liu, M., Tian, H., Zhuang, D., Zhang, Z., Zhang, W., Tang, X., and Deng, X.: Spatial and temporal patterns of China's cropland during 1990–2000: An analysis based on Landsat TM data, Remote Sensing of Environment, 98, 442-456, https://doi.org/10.1016/j.rse.2005.08.012, 2005.

**Specific comments:**

The number and quality of references in the first section of the Introduction is poor. I am sure there is more work done on LUCC changes that is more relevant to your work than done in Burkina Faso and Scandinavia. You can also leave these out as you mention more relevant ones later on.

Thanks for the comment. We have removed these two references and added two more relevant references in the first section of the introduction: Furthermore, Yu et al. (2020) found that the recent greening in China inferred a country-averaged surface cooling of 0.11 ℃. The study of Lin et al. (2020) showed that the urbanization tended to weak extreme precipitation events in urban agglomerations over coastal regions and intensify the influences on those in central/west China.

References:

Yu, L., Liu, Y., Liu, T., and Yan, F.: Impact of recent vegetation greening on temperature and precipitation over China, Agricultural and Forest Meteorology, 295, 10.1016/j.agrformet.2020.108197, 2020.

Lin, L., Gao, T., Luo, M., Ge, E., Yang, Y., Liu, Z., Zhao, Y., and Ning, G.: Contribution of urbanization to the changes in extreme climate events in urban agglomerations across China, Sci Total Environ, 744, 140264, https://doi.org/10.1016/j.scitotenv.2020.140264, 2020.

Adding a table to figure 4 with the percentages of LU classes would be more informative.

We agree with this comment and have added a table to figure 4 with the percentages of land use classes:

**Table 1. The percentages of land use and cover under four scenarios.**

| Scenarios | Cropland | Forest | Grassland | Water and wetland | Urban | Unused land |
|-----------|----------|--------|-----------|-------------------|-------|-------------|
| 1990 scenario | 29.15 | 42.82 | 23.50 | 1.65 | 0.19 | 2.69 |
| 2010 scenario | 28.48 | 43.60 | 23.13 | 1.79 | 0.86 | 2.14 |
| 20% scenario | 22.80 | 49.28 | 23.13 | 1.79 | 0.86 | 2.14 |
| 50% scenario | 14.58 | 57.50 | 23.13 | 1.79 | 0.86 | 2.14 |

How are the 32 vertical levels of the model spread? Are there enough layers near the bottom to trust the surface values you are evaluating such as skin surface temperature and 2-m relative humidity?

95

There were 32 eta levels of the model, and the top was at 50 hpa. We acknowledged that we did not test whether there were enough layers near the bottom to trust the surface values. However, there were many relevant studies which used similar or less vertical levels to study the changes of these surface variables (Hu et al., 2015; Yu et al., 2020). Moreover, Gallus et al. (2009) found that doubling the number of vertical levels from 31 to 62 did not result in a consistent improvement in the

100   precipitation forecasts and the skill might not be improved much by refining the number of levels. On the other hand, we acknowledge that the finding from Gallus's study may be different for different cases. However, refining/adding the number of levels will need much more computing resources and time to finish the simulations of such a big and nested domain in the study, which limits what we can achieve regarding it. So, we decided to leave this out in this study but may look at it in the future work. At the meantime, we will clarify it in the discussion part of the revised manuscript.

105

References:

Hu, Y., Zhang, X.-Z., Mao, R., Gong, D.-Y., Liu, H.-b., and Yang, J.: Modeled responses of summer climate to realistic land use/cover changes from the 1980s to the 2000s over eastern China, Journal of Geophysical Research: Atmospheres, 120, 167-179, https://doi.org/10.1002/2014jd022288, 2015.

110   Yu, L., Liu, Y., Liu, T., and Yan, F.: Impact of recent vegetation greening on temperature and precipitation over China, Agricultural and Forest Meteorology, 295, 10.1016/j.agrformet.2020.108197, 2020.

Gallus, W. A., Aligo, E. A., and Segal, M.: On the Impact of WRF Model Vertical Grid Resolution on Midwest Summer Rainfall Forecasts, Weather and Forecasting, 24, 575-594, 10.1175/2008waf2007101.1, 2009.

115   In Figure 5c (and others), why not show a qq-plot of model and observed rainfall instead? 50th percentile is not interesting to show and analyze.

Thanks for the comment. In the revised manuscript, we will show a qq-plot of observed and simulated rainfall instead of the 50th percentile rainfall (as shown as follows). We will also add the analysis of 99.95th percentile rainfall to further analyze

120   the changes of extreme rainfall.

[Figure]

Figure 5. The bias of (a) average summer rainfall (mm), (b) 99th percentile summer rainfall (mm/day) and (c) 50th percentile summer rainfall (mm/day) between the observed data and 2010 scenario, and (d) the qq-plot of observed rainfall versus simulated rainfall. The stippling regions show statistically significance of bias identified by t-test at a 5% significance level.

It seems urbanizations plays a role in the precipitation decrease between 1990 and 2010. Please consider using an urban scheme in WRF.

We acknowledge that the urbanization scheme may play a role in the WRF simulation for investigating the rainfall changes. However, it is difficult to re-run the simulation with urban scheme for this study, because it is very computing expensive and the time for running long-term simulations of such a big and nested domain is quite long. We will add this in the discussion of the revised manuscript and will take the urban scheme into consideration in future researches.

Why are the two areas (ALL-YRB) and (CTF-YRB) analyzed separately? Is there a rational in being interested in the converted areas specifically? Is analyzing more populated areas separately more interesting perhaps? As that is where the impact will be felt, not in the new forests.

140     Thanks for the comment. The reason we analyzed the two areas (ALL-YRB) and (CTF-YRB) separately was to investigate whether the land use changes at local scale influenced climate among the whole basin. We agree that analysing more populated areas separately is more interesting and will add the relevant results in revised manuscript.

    Line 275-277 please reconsider/rewrite.

145

    Thanks for the comment. We will revise the results according to all reviewers' comments and then rewrite Line 275-277.

---

## Author Comment (AC2) · 19 Jan 2021

**Reply to Referee comment 2**

Dear Editors and Reviewers:

We would like to thank the editor and all reviewers for their valuable suggestions and comments on the manuscript. These comments have not only improved the quality of the current manuscript, but also are beneficial to our future research in general. All point-by-point responses are presented as follows and we will carefully revise the manuscript based on these comments. For clarity, all comments are given in the original version, while responses are marked in blue.

**Anonymous Referee #2**

The manuscript by Li et al "Impacts of land use/cover change and reforestation on summer rainfall for the Yangtze River Basin" used the WRF model to investigate how land cover changes and reforestation affect summer rainfall. The research topic is important given the massive ecological projects in China and its climate impact is worth studying. The manuscript is generally well-written, but I also have major comments for the authors.

Thanks for the positive evaluations and comments, all comments and suggestions have been addressed and will be incorporated into the revised manuscript.

1. For WRF model simulation, how land cover changes were implemented in the model needs more detailed explanations as different land surface models have different representations of land cover. It is still unclear what surface conditions/variables had been modified for the Noah-MP model to correctly reflect the intended land cover changes. I also have questions about the rationality of the randomly chosen crops for the two restoration scenarios.

The land use changes were included in the WRF model by modifying the geographical static data used in the model which further changed the simulation of subprocesses such as the vegetation phenology, canopy stomatal resistance, runoff and groundwater in the land surface model Noah-MP (Li et al. 2018). Many parameters were used in Noah-MP to describe the characteristics of different land use types, such as albedo, HVT (Top of canopy), LAI (Monthly leaf area index), and VCMX25 (Maximum rate of carboxylation at 25 °C). When the land use changed, these parameters changed accordingly which finally led to the changes in substance and energy exchanges between atmosphere and land surface. The geographical static data we used in the WPS is *landuse_30s_with_lakes* which was download from the WRF website. The land use data of 1990 and 2010 scenarios were derived from the Landsat thematic mapper (TM) digital images. And then, in the YRB, we replaced the land use data from the *landuse_30s_with_lakes* with the land use data from the Landsat thematic mapper (TM) digital images.

Finally, we randomly changed 20% and 50% of the croplands to be forests using the 2010 scenario as a baseline to produce 20% and 50% reforestation scenarios. Both land cover data (downloaded from the WRF website and derived from the digital images) have a resolution of 1km. As the resolutions of outer and inner WRF domain were set as 75km and 15km, respectively, the post-processed land cover data was resampled from 1km to 75km and 15km by the WPS (the WRF Preprocessing System). The percentages of land cover under four scenarios after resampled to 15km are presented in Table A1 below. The dominant land use categories in model grids were used for the Noah-MP model to correctly reflect the intended land cover changes. We admit that the randomly chosen crops for the two restoration scenarios is a limitation in our study, as we have explained in the discussion section, the restoration processes usually happen in specific areas that are related to local policy. However, it is a challenge to gather all related policies from multiple local governments over such a big basin. It can be also noticed that the crops are mainly located in specific areas such as Sichuan Basin and the middle- and down-stream of the YRB. Although we chose the crop grids randomly, the restoration grids concentrated in these specific areas which was similar with the real processes.

**Table A1. The percentages of land use and cover under four scenarios after resampling.**

| Scenarios | Cropland | Forest | Grassland | Water and wetland | Urban | Unused land |
|---|---|---|---|---|---|---|
| 1990 scenario | 28.67 | 44.37 | 24.63 | 0.58 | 0.06 | 1.69 |
| 2010 scenario | 28.12 | 45.02 | 24.60 | 0.69 | 0.45 | 1.12 |
| 20% scenario | 22.97 | 51.55 | 24.83 | 0.69 | 0.54 | 1.12 |
| 50% scenario | 14.76 | 58.49 | 25.32 | 0.69 | 0.58 | 1.12 |

Reference:

Li, J., Chen, F., Zhang, G., Barlage, M., Gan, Y., Xin, Y., and Wang, C.: Impacts of Land Cover and Soil Texture Uncertainty on Land Model Simulations Over the Central Tibetan Plateau, Journal of Advances in Modeling Earth Systems, 10, 2121-2146, https://doi.org/10.1029/2018ms001377, 2018.

2. When comparing simulation results between different experiments, the authors need to conduct statistical significance tests to determine whether the signal is robust while excluding any noise and random changes which may lead to misinterpretation.

We agree with this comment. We have conducted statistical significance tests to determine whether the signal is robust when comparing simulation results between different experiments. We display the revised Fig.5 and Fig. 6 as follows; other results of significance test will be incorporated into the revised manuscript.

[Figure]

60

**Figure 5. The bias of (a) average summer rainfall (mm), (b) 99th percentile summer rainfall (mm/day) and (c) 50th percentile summer rainfall (mm/day) between the observed data and 2010 scenario, and (d) the qq-plot of observed rainfall versus simulated rainfall. The stippling regions show statistically significance of bias identified by t-test at a 5% significance level.**

65

[Figure]

**Figure 6. The changes in (a) average summer rainfall (mm), (b) 99th percentile summer rainfall (mm/day) and (c) 50th percentile summer rainfall (mm/day) between the 1990 scenario and 2010 scenario. The stippling regions show statistically significance of changes identified by t-test at a 5% significance level.**

70

3. I hope the authors could provide more mechanistic explanations of the results. For example, why did the 20% reforestation result in more precipitation changes than the 50% reforestation scenario?

Thanks for the comment. We will provide more mechanistic explanations of the results in the revised manuscript. As for the
20% reforestation resulted in more precipitation changes than the 50% reforestation scenario, after analysing the changes in
the water vapor mixing ratio at 2m and upward moisture flux at the surface (Fig. A2 and A3), we found that the number of
grids showing increased upward moisture flux in the 50% scenario slightly exceeded that in the 20% scenario. In contrast, the
2m water vapor mixing ratio increased over almost all basin in the 20% scenario while showed large decreases in the midstream
of the basin in the 50% scenarios. From the surface level to the 2m level, the moisture kept increased in the 20% scenarios
while decreased in the 50% scenarios. This suggested that the distribution of moisture may be changed by the horizontal
transportation process. Moreover, Yu et al. (2020) found that the vegetation greening reduced rainfall in some region in the
southern China which may be caused by the East Asian monsoon. As the East Asian monsoon significantly influenced the
summer precipitation patterns in China (Ding et al., 2007). All these information and explanations will be incorporated into
the revised manuscript.

Ding, Y., Ren, G., Zhao, Z., Xu, Y., Luo, Y., Li, Q., and Zhang, J.: Detection, causes and projection of climate change over
China: An overview of recent progress, Advances in Atmospheric Sciences, 24, 954-971, https://doi.org/10.1007/s00376-007-
0954-4, 2007.
Yu, L., Liu, Y., Liu, T., and Yan, F.: Impact of recent vegetation greening on temperature and precipitation over China,
Agricultural and Forest Meteorology, 295, https://doi.org/10.1016/j.agrformet.2020.108197, 2020.

[Figure]

**Figure A2. The changes in (a-b) 2m water vapor mixing ratio (g/kg) between the 20% scenario and 2010 scenario, and
between the 50% scenario and 2010 scenario. The stippling regions show statistically significance of changes identified
by t-test at a 5% significance level.**

[Figure]

**Figure A3. The changes in (a-b) upward moisture flux at the surface (kg/m$^2$) between the 20% scenario and 2010 scenario, and between the 50% scenario and 2010 scenario. The stippling regions show statistically significance of changes identified by t-test at a 5% significance level.**

**Specific comments:**

L9: There is another terminology "Grain for Green" frequently used in the literature for "Returning Farmland to Forest Program". Which one is better acknowledged?

Thanks for the comment. Both the terminologies are correct, and the "Grain for Green" may be more widely used. We will change all the "Returning Farmland to Forest Program" to "Grain for Green" in the revised manuscript.

L130-140: What kinds of WRF experiments have been conducted to compare different schemes/parameterizations, what domain and simulation length was used for the comparison experiments?

According to previous studies in China (e.g., Hu et al., 2015; Zhang et al., 2019; Feng et al., 2012; Xue et al., 2017), we chose three microphysical schemes (i.e., Purdue Lin Scheme (Lin), WRF Single-moment 5-class Scheme (WSM5), and Eta Scheme (Ferrier)) and two cumulus parameterization (i.e., Kain-Fritsch Scheme (KFN) and Grell–Devenyi Ensemble Scheme (GD)) for tests. Five parameterization scheme combinations (i.e., Lin-KFN, WSM5-KFN, Ferrier-KFN, Lin-GD and WSM5-GD) were used to simulate the rainfall and temperature for the Yangtze River basin during 2005 summer, as there were several rainstorm events in 2005 summer for this basin. The most suitable parameterization schemes were chosen by comparing the performance of these five combinations. The domain setting was same as the whole experiment which can be seen in Fig. 2. The simulation length was 3 months from June to August. We will add these information and explanations in the method and results of the revised manuscript.

References:

Feng, J.-M., Wang, Y.-L., Ma, Z.-G., and Liu, Y.-H.: Simulating the Regional Impacts of Urbanization and Anthropogenic Heat Release on Climate across China, Journal of Climate, 25, 7187-7203, https://doi.org/10.1175/JCLI-D-11-00333.1, 2012.

125 Hu, Y., Zhang, X.-Z., Mao, R., Gong, D.-Y., Liu, H.-b., and Yang, J.: Modeled responses of summer climate to realistic land use/cover changes from the 1980s to the 2000s over eastern China, Journal of Geophysical Research: Atmospheres, 120, 167-179, https://doi.org/10.1002/2014jd022288, 2015.

Xue, H., Jin, Q., Yi, B., Mullendore, G. L., Zheng, X., and Jin, H.: Modulation of Soil Initial State on WRF Model Performance Over China, Journal of Geophysical Research: Atmospheres, 122, 11,278-211,300, https://doi.org/10.1002/2017JD027023, 130 2017.

Zhang, H., Wu, C., Chen, W., and Huang, G.: Effect of urban expansion on summer rainfall in the Pearl River Delta, South China, Journal of Hydrology, 568, 747-757, https://doi.org/10.1016/j.jhydrol.2018.11.036, 2019b.

L145-149: It is better to also report the quantities of land cover changes between 1990 and 2010.

135

Thanks for the comment. We have added a table with the quantities of land cover under four scenarios. The quantities of land cover changes between 1990 and 2010 can be easily found from this table.

**Table 1. The percentages of land use and cover under four scenarios.**

| Scenarios | Cropland | Forest | Grassland | Water and wetland | Urban | Unused land |
|---|---|---|---|---|---|---|
| 1990 scenario | 29.15 | 42.82 | 23.50 | 1.65 | 0.19 | 2.69 |
| 2010 scenario | 28.48 | 43.60 | 23.13 | 1.79 | 0.86 | 2.14 |
| 20% scenario | 22.80 | 49.28 | 23.13 | 1.79 | 0.86 | 2.14 |
| 50% scenario | 14.58 | 57.50 | 23.13 | 1.79 | 0.86 | 2.14 |

140

L148: How did the random changes from cropland to forest being incorporated in the model surface land condition at 15 km resolution? I am not sure whether this choice is necessary. What land variables had been modified to represent the land cover change in WRF model and what are their changes? What types of forest were used in the reforestation experiment? How many 145 grid boxes experienced land cover change?

The land use changes were included in the WRF model by modifying the geographical static data used in the model which further changed the simulation of subprocesses such as the vegetation phenology, canopy stomatal resistance, runoff and groundwater in the land surface model Noah-MP (Li et al. 2018). Many parameters were used in Noah-MP to describe the

150 characteristics of different land use types, such as albedo, HVT (Top of canopy), LAI (Monthly leaf area index), and VCMX25 (Maximum rate of carboxylation at 25 °C). When the land use changed, these parameters changed accordingly which finally led to the changes in substance and energy exchanges between atmosphere and land surface. The geographical static data we used in the WPS is *landuse_30s_with_lakes* which was download from the WRF website. The land use data of 1990 and 2010 scenarios were derived from the Landsat thematic mapper (TM) digital images. And then, in the YRB, we replaced the land

155 use data from the *landuse_30s_with_lakes* with the land use data from the Landsat thematic mapper (TM) digital images. Finally, we randomly changed 20% and 50% of the croplands to be forests using the 2010 scenario as a baseline to produce 20% and 50% reforestation scenarios. Both the land use data (downloaded from the WRF website and derived from the digital images) have a resolution of 1km. As the resolution of inner domain of the WRF model was set as 15km, the post-processed land cover data were resampled from 1km to 15km by the WPS (the WRF Preprocessing System). Then, the dominant land

160 use categories in model grids were used for the Noah-MP model to correctly reflect the intended land cover changes. There were two main types of croplands, i.e., dry cropland and pasture (USGS code 2), and irrigated cropland and pasture (USGS code 3), and three main types of forest, i.e., shrubland (USGS code 8), savanna (USGS code 10) and deciduous broadleaf forest (USGS code 11). For the 20% and 50% scenarios, there were 408 and 1060 cropland grids experienced land cover changes, while the total grids of cropland in the 2010 scenarios was 2231.

165

Reference:

Li, J., Chen, F., Zhang, G., Barlage, M., Gan, Y., Xin, Y., and Wang, C.: Impacts of Land Cover and Soil Texture Uncertainty on Land Model Simulations Over the Central Tibetan Plateau, Journal of Advances in Modeling Earth Systems, 10, 2121-2146, https://doi.org/10.1029/2018ms001377, 2018.

170

L170: What about statistical significance levels of these precipitation changes? This needs to be reported for this and other figures as well.

Thanks for the comment. We have conducted statistical significance tests with t-test at 5% significance level for all spatial

175 difference plots to determine whether the signals are robust when comparing simulation results between different experiments. We display the revised Fig. 6 as follow; other significance test results will be incorporated into the revised manuscript.

[Figure]

**Figure 6. The changes in (a) average summer rainfall (mm), (b) 99th percentile summer rainfall (mm/day) and (c) 50th percentile summer rainfall (mm/day) between the 1990 scenario and 2010 scenario. The stippling regions show statistically significance of changes identified by t-test at a 5% significance level.**

L224: Why did 20% and 50% reforestation grids at the model resolution are different?

For the 20% reforestation scenario, only 20% cropland grids of the 2010 scenario were changed to forest grids. While for the 50% reforestation scenario, the proportion of cropland grids changed to forest grids was 50%. Moreover, the two reforestation scenarios were independently produced using random sampling. Thus, the 20% and 50% reforestation grids are different. This will be clarified in the revised manuscript.

L241: For this section, the overall decreases in both LHF and SHF after reforestation were unexpected to me. Not sure if these changes are robust enough. Typical, ET would increase after reforestation, as described in the introduction, so how to explain this result?

Thanks for the comment. From the results of significance test in the revised Fig. 12, we found that the increases of LHF were more significant than decreases after reforestation. Moreover, we added a quantitative investigation on the changes in LHF and SHF over the whole basin and found that the multiyear average summer daily LHF increased by $2.08 \times 10^3$ and $4.82 \times 10^3$ $W/m^2$ for the 20% and 50% scenarios, respectively, while the multiyear average summer daily SHF decreased by $4.30 \times 10^3$ and increased by $4.25 \times 10^3$ $W/m^2$ for the 20% and 50% scenarios, respectively. Therefore, the ET did increases after reforestation. We will add these results in the revised manuscript.

[Figure]

**Figure 12. The changes in (a-b) latent heat flux (LHF, W/m2), (c-d) sensible heat flux (SHF, W/m2) and (e-f) PBL height (PBLH, m) between the 20% scenario and 2010 scenario, and between the 50% scenario and 2010 scenario. The stippling regions show statistically significance of changes identified by t-test at a 5% significance level.**

205

L259: What about the changes in near-surface air temperature? For example, 2m air temperature.

Thanks for the comment. We actually analysed the changes in 2m air temperature which were not showed in the manuscript. And the results were almost the same as the changes in surface skin temperature. We display the changes in 2m air temperature

210 in Figure A1 below. Considering that the length of the paper is too long, we will show it in the appendix.

[Figure]

**Figure A1. The changes in (a-b) surface skin temperature (℃) and (c-d) 2m air temperature (℃) between the 20% scenario and 2010 scenario, and between the 50% scenario and 2010 scenario. The stippling regions show statistically significance of changes identified by t-test at a 5% significance level.**

L276-277: Any evidence to support this argument, given the latent heat flux decreased?

We have done the quantitively analysis of the changes of LHF over the whole basin, and have found that the latent heat flux increases after reforestation. We will revise this part of results and then rewrite this argument.

L296-297: How many urban grids had changed between 1990 and 2010? Whether urban expansion will affect the entire Yangtze river basin?

There were 32 urban grids out of the total of 7935 grids in Yangtze river basin, had been changed between 1990 and 2010. Moreover, as the urban expansion mainly concentrated in the midstream and downstream of Yangtze River basin, it was

difficult to identify how much impact the urban expansion would have on the Yangtze river basin through only 32 grids, probably be negligible. We assume that urbanization mainly affects the local region.

230    L332: Is there actual data to support the increased water vapor mixing?

From the model data, it can be found that the water vapor mixing increased at the 2m, especially for the 20% scenario. For the 50% scenario, areas with the significant water vapor mixing ratio increased were more than areas with significant water vapor mixing ratio decreased. We will add spatial difference plots in the revise manuscript to show this result.

235

**Figure A2. The changes in (a-b) 2m water vapor mixing ratio (g/kg) between the 20% scenario and 2010 scenario, and between the 50% scenario and 2010 scenario. The stippling regions show statistically significance of changes identified by t-test at a 5% significance level.**

240

L335: Why is the precipitation response larger in 20% than in the 50% scenario? There is no related explanation or discussion.

After analysing the changes in the water vapor mixing ratio at 2m and upward moisture flux at the surface (Fig. A2 and A3), we found that the number of grids showing increased upward moisture flux in the 50% scenario slightly exceeded that in the 245    20% scenario. In contrast, the 2m water vapor mixing ratio increased over almost all basin in the 20% scenario while showed large decreases in the midstream of the basin in the 50% scenarios. From the surface level to the 2m level, the moisture kept increased in the 20% scenarios while decreased in the 50% scenarios. This suggested that the distribution of moisture may be changed by the horizontal transportation process. Moreover, Yu et al. (2020) found that the vegetation greening reduced rainfall in some region in the southern China which may be caused by the East Asian monsoon. As the East Asian monsoon 250    significantly influenced the summer precipitation patterns in China (Ding et al., 2007). All these information and explanations will be incorporated into the revised manuscript.

Ding, Y., Ren, G., Zhao, Z., Xu, Y., Luo, Y., Li, Q., and Zhang, J.: Detection, causes and projection of climate change over China: An overview of recent progress, Advances in Atmospheric Sciences, 24, 954-971, https://doi.org/10.1007/s00376-007-0954-4, 2007.

Yu, L., Liu, Y., Liu, T., and Yan, F.: Impact of recent vegetation greening on temperature and precipitation over China, Agricultural and Forest Meteorology, 295, https://doi.org/10.1016/j.agrformet.2020.108197, 2020.

[Figure]

**Figure A2. The changes in (a-b) 2m water vapor mixing ratio (g/kg) between the 20% scenario and 2010 scenario, and between the 50% scenario and 2010 scenario. The stippling regions show statistically significance of changes identified by t-test at a 5% significance level.**

[Figure]

**Figure A3. The changes in (a-b) upward moisture flux at the surface (kg/m$^2$) between the 20% scenario and 2010 scenario, and between the 50% scenario and 2010 scenario. The stippling regions show statistically significance of changes identified by t-test at a 5% significance level.**

---

## Author Comment (AC3) · 19 Jan 2021

**Reply to Referee comment 3**

Dear Editors and Reviewers:

We would like to thank the editor and all reviewers for their valuable suggestions and comments on the manuscript. These comments have not only improved the quality of the current manuscript, but also are beneficial to our future research in general. All point-by-point responses are presented as follows and we will carefully revise the manuscript based on these comments. For clarity, all comments are given in the original version, while responses are marked in blue.

**Anonymous Referee #3**

The authors of the manuscript "Impacts of land use/cover change and reforestation on summer rainfall for the Yangtze River Basin" present work that show the effects of land use and land cover change on regional climate processes including summer rainfall. The manuscript shows the importance of better understanding these effects and has some interesting discussion points. These types of studies are difficult to do and this is a great start. However, in my opinion, the points outlined in this review need to be addressed for this work to have scientific merit.

Thanks for the positive evaluations and comments, all the comments and suggestions have been addressed and will be incorporated into the revised manuscript.

**General comments:**

1. The methods used to change land cover need to be discussed further as other reviewers have mentioned. The land surface model (Noah-MP) is complex and offers many options to better represent land surface processes. The land surface model is only mentioned once in the text. Noah-MP contains too many options that need to be carefully chosen for this to be glossed over. Additionally, Noah-MP uses only the dominant land use category when calculating surface fluxes, so at 15km an increase in forest will not matter if it doesn't become the dominant category. This may help explain the inconsistent results between the 20% and 50% reforestation but without more information, it's hard to say.

Thanks for the comment. The land use changes were included in the WRF model by modifying the geographical static data used in the model which further changed the simulation of subprocesses such as the vegetation phenology, canopy stomatal resistance, runoff and groundwater in the land surface model Noah-MP (Li et al. 2018). Many parameters were used in Noah-MP to describe the characteristics of different land use types, such as albedo, HVT (Top of canopy), LAI (Monthly leaf area index), and VCMX25 (Maximum rate of carboxylation at 25 °C). When the land use changed, these parameters changed

accordingly which finally led to the changes in substance and energy exchanges between atmosphere and land surface. The geographical static data we used in the WPS is *landuse_30s_with_lakes* which was download from the WRF website. The land use data of 1990 and 2010 scenarios were derived from the Landsat thematic mapper (TM) digital images. And then, in the YRB, we replaced the land use data from the *landuse_30s_with_lakes* with the land use data from the Landsat thematic mapper (TM) digital images. Finally, we randomly changed 20% and 50% of the croplands to be forests using the 2010 scenario as a baseline to produce 20% and 50% reforestation scenarios. Both the land cover data (downloaded from the WRF website and derived from the digital images) have a resolution of 1km. As the resolution of inner domain of WRF model was set as 15km, the post-processed land cover data was resampled from 1km to 15km by the WPS (the WRF Preprocessing System). Then, the dominant land cover categories in model grids were used for the Noah-MP model to correctly reflect the intended land cover changes. For 20% and 50% scenarios, there were 408 and 1060 cropland grids experience land cover changes, while the total grids of cropland in 2010 scenarios was 2231.

Reference:

Li, J., Chen, F., Zhang, G., Barlage, M., Gan, Y., Xin, Y., and Wang, C.: Impacts of Land Cover and Soil Texture Uncertainty on Land Model Simulations Over the Central Tibetan Plateau, Journal of Advances in Modeling Earth Systems, 10, 2121-2146, https://doi.org/10.1029/2018ms001377, 2018.

2. The limitations of using a convective parameterization when investigating rainfall extremes needs to be discussed. In a region with large vertical relief, the choice to use a course resolution for this study should be justified. Convection permitting scales (<4km) not only allow for better representation of precipitation processes, but also better land surface representation (including topography).

Thanks for the comment. We realize that convective parameterizations differ greatly in their treatment of the cloud up draughts and down draughts, mass-flux closure and triggering, often assuming that one is averaging over both cloud up draughts and the subsiding environment. As a result, all these schemes are better at predicting the area-average rainfall (Clark et al., 2016). Additionally, the cumulus parameterizations also introduce uncertainties to the model results (Liu et al., 2017). We also agree that higher model resolution can better represent precipitation processes and land surface. However, this study focused on the Yangtze River basin, which had a total area of $\sim 1.8 \times 10^6$ km$^2$. Considering the huge area, the spatial resolution of 15 km may be enough to reproduce reliable rainfall patterns. In some other studies which evaluated the impacts of land use/cover changes on climate over such a big region, the resolution of model was usually even coarser (e.g., Hu et al., 2014; Zhang et al., 2017). We will add above clarification and the relevant information and discussion in the Discussion section of the revised manuscript.

References:

Clark, P., Roberts, N., Lean, H., Ballard, S. P., and Charlton-Perez, C.: Convection-permitting models: a step-change in rainfall forecasting, Meteorological Applications, 23, 165-181, https://doi.org/10.1002/met.1538, 2016.

Liu, C., Ikeda, K., Rasmussen, R., Barlage, M., Newman, A. J., Prein, A. F., Chen, F., Chen, L., Clark, M., Dai, A., Dudhia, J., Eidhammer, T., Gochis, D., Gutmann, E., Kurkute, S., Li, Y., Thompson, G., and Yates, D.: Continental-scale convection-
70  permitting modeling of the current and future climate of North America, Climate Dynamics, 49, 71-95, https://doi.org/10.1007/s00382-016-3327-9, 2016.

Hu, Y., Zhang, X.-Z., Mao, R., Gong, D.-Y., Liu, H.-b., and Yang, J.: Modeled responses of summer climate to realistic land use/cover changes from the 1980s to the 2000s over eastern China, Journal of Geophysical Research: Atmospheres, 120, 167-179, https://doi.org/10.1002/2014jd022288, 2015.

75  Zhang, X., Xiong, Z., Zhang, X., Shi, Y., Liu, J., Shao, Q., and Yan, X.: Simulation of the climatic effects of land use/land cover changes in eastern China using multi-model ensembles, Global and Planetary Change, 154, 1-9, https://doi.org/10.1016/j.gloplacha.2017.05.003, 2017.

3. The model validation is insufficient. Look to Liu et al., 2017, for an example of full model validation. To be specific, I
80  would like to see the figures reworked to show the spatial patterns of rainfall on a seasonal and annual basis in the observations and in the control simulations. Furthermore, the figures should include a representation of percent change in rainfall. A bias of 600mm of rainfall during the summer months is a lot if the average summer rainfall is only 1000mm. This information isn't shown so it's hard to know if the bias is significant. Statistical testing should also be included where appropriate. Additionally, validation of other climatic components that contribute to rainfall (such as the vertical structure of the atmosphere, PBLH,
85  CAPE, CIN) would aid this study. Validation of surface fluxes would also help build a better picture of how well the model can represent this region. There are several eddy-covariance towers in the eastern part of the domain and a comparison of sensible and latent heat flux to those towers would be interesting. Any change that is presented should have an accompanying discussion of validation for that component. Showing Figure 10 but compared to observations would be necessary to see if WRF can capture extreme rainfall.

90

Thanks for the comments. We will add the figures to show the spatial patterns of rainfall on a seasonal and annual basis for the observations and in the control simulations. The figures will include a representation of percent change and the results of statistical testing in the revised manuscript. In addition, we are aware of that the validation of other climate components and surface fluxes can be helpful. Actually, we have already looked for the data from the several eddy-covariance towers in the
95  eastern part of the domain that reviewer mentioned here. However, we can only get flux data of one of these towers from the China Nation Science and Technology Infrastructure (http://www.cnern.org.cn/index.jsp) and the data are from 2003 to 2010 which is mismatch with our simulation period. So unfortunately, we don't have such observation data in the study. In this case, we will add ERA5 dataset in the revised manuscript for a further model evaluation, including the 850hpa humidity and surface fluxes variables, such as sensible and latent heat fluxes. Besides, we will also add the temperature evaluation based on the

observed temperature data, which is the only available observation data we have besides observed precipitation in the study. Furthermore, we will also show the probability distribution functions of rainfall in 2010 scenarios compared to observations to see if WRF can capture extreme rainfall.

4. The taylor diagrams are honestly pretty confusing, I would remove them and provide a table of biases instead. The correlation coefficients are rather low for temperature (the easiest for the model to accurately capture) and lower for rainfall when compared to observations. This leads me to believe that the model isn't configured properly for this region. If the above issues were tackled, then this opinion might change. One way to show that the model is well validated is to show that the temperature and rainfall falls within the spread of observations. Comparison to not only the station data but to an independent gridded dataset (such as ERA5, CRU, etc.) would strengthen this point.

Thanks for the comment. We have removed the taylor diagrams. The correlation coefficient was rather low for temperature might be because that the taylor diagrams were calculated at station basis by interpolating the simulation results to stations. When calculating the correlation coefficient of temperature at grid scale, the result was acceptable. We have added qq-plots of temperature and rainfall between simulated and observed data (Fig. 4 and Fig. 5), and find that the distribution of temperature and rainfall simulated by model are linear correlated with those of observation. We will also compare the simulated data with both station data and ERA5 dataset to see whether the simulated temperature and rainfall fall within the spread of observations.

[Figure]

**Figure 4. The bias of (a) average summer temperature (°C), and (b) the qq-plot of observed temperature versus simulated temperature. The stippling regions show statistically significance of bias identified by t-test at a 5% significance level.**

[Figure]

**Figure 5. The bias of (a) average summer rainfall (mm), (b) 99th percentile summer rainfall (mm/day) and (c) 50th percentile summer rainfall (mm/day) between the observed data and 2010 scenario, and (d) the qq-plot of observed rainfall versus simulated rainfall. The stippling regions show statistically significance of bias identified by t-test at a 5% significance level.**

5. The percentiles of rainfall need to be defined better. What does 99th percentile mean in this case? Is it the 99th percentile of rainfall events over the 11 years? Without sub-daily rainfall, I'm not sure that this qualifies as extreme per se. A common extreme rainfall metric is the 99th percentile of daily maximum rainfall (requires sub daily rainfall to properly calculate). In my country, the storms that produce flash flooding often last only a few hours, vs a monsoon type rain that produces flooding from many, many hours of low intensity rainfall. More discussion of rainfall in this region would put this information into context. I would remove the figures that show changes to median rainfall and instead discuss some other metric of interest.

Thanks for the comment. The 99th percentile is the multiyear average value from the 99th percentile rainfall in each year. We did not use sub-daily rainfall because the flash flooding in Yangtze River basin was often caused by continuous rainfall last for a few days, as it is a big basin. A few hours of high intensity rainfall would not cause severe flooding due to the construction of cascade reservoirs along the river. Moreover, we have replaced the results of median rainfall with 99.95th rainfall in order to give a more comprehensive assess on the different levels of extreme rainfall. The analysis and relevant results will be added in the revised manuscript.

6. All the figures showing change between simulations need to have statistical testing. The figures all look very noisy and some of the changes to precipitation could be because the storms moved, not because more rain fell.

We agree with this comment. We have conducted statistical tests and modified all the figures showing change between simulations to present the results of statistical tests. We display the revised Fig. 7 as follows; results of other significance test will be incorporated into the revised manuscript.

[Figure]

Figure 7. The changes in (a-b) average summer rainfall (mm), (c-d) 99th percentile summer rainfall (mm/day) and (e-f) 50th percentile summer rainfall (mm/day) between the 20% scenario and 2010 scenario, and between the 50% scenario and 2010 scenario. The stippling regions show statistically significance of bias identified by t-test at a 5% significance level.

7. Instead of bar graphs, boxplots or violin plots should be shown. This will capture the distribution of the change.

Thanks for the comment. We will add boxplots to show the distribution of the changes in the revised manuscript.

**Minor specific comments:**

The convention I have seen for abbreviating land use and land cover change is LULCC not LUCC.

Thanks for the comment. We will replace all the "LUCC" with "LULCC" in the revised manuscript.

165    There are some English language errors in the text, but these don't bother me that much and have been covered by other reviewers.

We will carefully check the whole paper to improve the quality.

---

## Author Response (AR1)

**Reply to Editor Decision**

Comments to the Author: Dear Wei Li and co-author,

**5**

Thank you for posting your responses to the three referees' reports. The reviewers raised some important comments and suggestions - especially considering the model setup and validation, the reforestation scenarios and the interpretation of the results. From reading your responses, I can see that you seriously considered their critiques, which I have confidence that will improve the quality of the manuscript. In addition to the comments made by the reviewers, I would like to suggest another

- 10 analysis that I believe will contribute to the scientific quality of the paper: many recent studies tie between rainfall intensification, air temperature and humidity increase (i.e. the Clausius–Clapeyron relation); in your work, I see the potential of exploring how this relation is changing following the change in vegetation cover. Please consider this point, as it might add another interesting scientific perspective to your work. Based on my reading of the original manuscript and your replies to the referees, I find this to be a potentially interesting paper that might fit the scope of HESS and could be of interest to the
- 15 hydrological community. Therefore, I invite you to upload a revised manuscript, incorporating the proposed changes and additions, and making any other modifications where you see fit. In your response, please provide a point-to-point answer to the comments made by the reviews, and a track-changed version of the manuscript. I look forward to receiving the revised manuscript.
- 20 Sincerely,

Nadav Peleg

**Dear Editor:**

- 25 We would like to appreciate the editor's and all reviewers' valuable suggestions and comments on the manuscript. These comments have not only improved the quality of the current manuscript but also are beneficial to our future research in general. All point-by-point responses are presented in our replies and we have carefully revised the manuscript based on these comments. Moreover, considering the editor's suggestions in the comment, we have done some analysis and here is the reply:
- 30 We try to find the relations between rainfall and temperature under different scenarios using the linear regression method, which is recommended in the previous study (Zhou et al., 2016). Then, whether and how the relation between rainfall and temperature changes following the change in vegetation cover is explored by comparing the regression coefficients. First, the

average daily rainfall and temperature for the 2010 scenario and two hypothetical reforestation scenarios (20% scenario and 50% scenario) are calculated for each year at the grid-scale. Considering the simulation period is from 2001 to 2010, there are

- 35 10 values for rainfall and temperature for each grid, respectively. The relation between rainfall and temperature is then established using the linear regression method for each grid cell. Fig. R1 shows the spatial distributions and boxplots of the regression coefficients in terms of the percentage. From the figures, it can be seen that there are not many differences in the regression coefficients among the three scenarios. Some reasons may explain this result. (1) There are only 10 points for each grid to calculate the regression coefficient, which may bring large uncertainties. It is hard to determine whether the regression
- 40 equation can well represent the relation between rainfall and temperature as the correlation between rainfall and temperature is insignificant at a 5% significance level in many places. (2) Although the vegetation cover changes, the rainfall and temperature among ten years are not monotonic, which means no apparent trends for rainfall and temperature. That is why the regression coefficients between rainfall and temperature are close to zero for most grids. (3) The simulation biases further enlarge the uncertainties of the relation between rainfall and temperature. From the analyses above, we do not find the change
- 45 in the relation between rainfall and temperature in terms of the vegetation cover change in this study. We would like not to include this result in the revised manuscript regarding the length of revised manuscript, which is already pretty long (13 figures and 6 figures in appendix), although we think this is a useful testing and checking. Thank you all the same for this comment and suggestion.
- 50 References:

Zhou, Y., Luo, M., and Leung, Y.: On the detection of precipitation dependence on temperature, Geophysical Research Letters, 43, 4555-4565, https://doi.org/10.1002/2016gl068811, 2016.

55 Figure R1. The spatial distributions and boxplots of the regression coefficients (%) for the 2010 scenario and two hypothetical reforestation scenarios. The stippling regions show statistically significance of changes identified by t-test at a 5% significance level.

60

Dear Editors and Reviewers:

We would like to thank the editor and all reviewers for their valuable suggestions and comments on the manuscript. These comments have not only improved the quality of the current manuscript, but also are beneficial to our future research in general.

65 All point-by-point responses are presented as follows and we have carefully revised the manuscript based on these comments. For clarity, all comments are given in the original version, while responses are marked in blue.

**Emma Daniels (Referee)**

**General comments:**

- 70 In general, I think the paper has interesting results and could be published. However, the quality of English needs to be improved in some parts (mainly abstract, introduction and methods). Moreover, I miss details in your method such as your definition of summer (i.e. which months are analyzed) and details on the land use maps (e.g. a table with percentages) and how they are included. I think the paper would benefit from analysis of an additional parameter for extreme precipitation, such as rainfall above the 90th percent as with 10 years of data (i.e. 900 data points assuming a summer of 3 months) the 99th percentile
- 75 alone might be misleading. Furthermore, I miss an explanation why precipitation is thought to increase with further reforestation but decreased between 1990 and 2010 though forest cover increased. Also, I wonder why Shrubland (USGS code 8) and Savanna (USGS code 10) are chosen as a type of forest? Judging from the LANDUSE.TBL these classes are much more similar to Cropland and Pasture than forest, so I wonder if expanding these makes a difference or if you are mainly looking at the effect of the additional Broadleaf forest. I think the figures need work and should become more informative than 80 mainly harplets and spatial difference plots.
- 80 mainly barplots and spatial difference plots.

Thanks for the comments. We are sorry for the grammar problems in the manuscript. The manuscript has been proofread by a native English speaker. We have added the definition of summer in the introduction, and the summer defined in this manuscript is from June to August. Table 1 has been added in the revised manuscript to explain the percentages of different land use types in the whole herin. Moreover, the land use sheaver man included in the WDE model have added here experiment is characterized.

- 85 in the whole basin. Moreover, the land use changes were included in the WRF model by modifying the geographical static data used in the model which further changed the simulation of subprocesses such as the vegetation phenology, canopy stomatal resistance, runoff and groundwater in the land surface model Noah-MP (Li et al. 2018). Many parameters were used in Noah-MP to describe the characteristics of different land use types, such as albedo, HVT (Top of canopy), LAI (Monthly leaf area index), and VCMX25 (Maximum rate of carboxylation at 25 °C). When the land use changed, these parameters changed
- 90 accordingly which finally led to the changes in substance and energy exchanges between atmosphere and land surface. In the study, we used the U.S. Geological Survey (USGS) land cover with 30s resolution (~ 1km resolution;

"landuse\_30s\_with\_lakes") in the WRF Preprocessing System (WPS). The new land use data of 1990 and 2010 derived from the Landsat TM digital images at 1km resolution, was then used to replace the USGS land cover data in the WRF simulation in YRB. Finally, we randomly changed 20% and 50% of the croplands to be forests using the 2010 scenario as a baseline to

95 produce 20% and 50% reforestation scenarios.

- Given the comments from other reviewers, the 99.95th percent summer rainfall has been chosen to further analyze the extreme rainfall. Furthermore, the land use changes from 1990 to 2010 were not only attributed to the increase of forests, but also the change of other land uses. Therefore, although the forests increased between 1990 and 2010, the precipitation decreased with the joint impacts of all other land use changes.
- 100 Moreover, the land use categories of the 1990 and 2010 land use data from Landsat TM digital images were defined by Liu et al. (2002, 2005), which were commonly used in China; while, the USGS data for WRF modelling has 24 land use categories (including lake). Thus, we used the method of land use type conversions based on the study of Hu et al. (2015). According to this method, the four classes of land use in the Liu's category from Landsat TM digital images, including the Forest (Liu code 21), Shrub (Liu code 22), Sparse woodland (Liu code 23), and Cut over land (Liu code 24), were converted to four classes of
- 105 USGS land use category, including the Deciduous broadleaf forest (USGS code 11), Shrubland (USGS 45 code 8), Savanna (USGS code 10), and Savanna (USGS code 10), respectively. That was why Shrubland (USGS code 8) and Savanna (USGS code 10) were chosen as a type of forest.

All above information and more clarifications have been added in the method section of the revised manuscript. The figures in the revised manuscript have been improved, and we have also added more informative figures such as qq-plot, boxplots and

110 significance test in the revised manuscript. Please see the revised Fig. 4 and Fig. 10 as follows; other revised figures can be found in the revised manuscript.

---

## Referee Report (RR1)

My comments on the manuscript "Impacts of land use/cover change and reforestation on summer rainfall for the Yangtze River Basin" have largely been handled.

There are still English mistakes, but it doesn't affect the readability of text much.

I do believe that the paper would be more impactful if the authors made more of an attempt to understand the differences between the 20% forest cover scenario and the 50% forest cover scenario. Why would the 20% scenario produce larger changes than the 50% scenario? The surface fluxes look similar across the different scenarios, but the surface temperature and relative humidity are quite different (and even opposite in sign). The authors allude to perhaps changes to horizontal wind, but this would be very interesting if they chose to discuss this further. The authors might look to Eiras-Barca et al., 2020 (doi: 10.1111/nyas.14364) for inspiration as this study looks at deforestation and assesses moisture transport and surface roughness changes as a result.

---

## Author Response (AR2)

**Reply to Editor Decision**

Comments to the Author:

Dear Wei Li and co-authors,

I have now received the reports of the three referees. One referee suggested accepting the paper, another suggested minor revisions and another suggested major revisions. Reading their comments and suggestions and the text once again, I believe the manuscript needs another round of major revisions before it can be considered for publication in HESS.

Please consider carefully the following points in your revised manuscript and replies to the reviewers:

1. Justify the use of the 99.95th percentile of rainfall in your analysis.

2. Better explain the differences between the 20% forest cover scenario and the 50% forest cover scenario (especially address the comment made by the third reviewer - why would the 20% scenario produce larger changes than the 50% scenario?).

3. Why temperature was used for the comparison? Also, comment on what seems to be a systematic bias in the temperature.

4. What land cover changes happened between 1990 and 2010?

5. Explain the contradiction in LHF values between Fig. 13 and the text.

I summarized the above points based on the reports of the referees. Please, in your replies, address each of the points raised in their reports (not only the points summarized above). Also, consider having a native English speaker proofreading the manuscript. I am looking forward to receiving your revised manuscript.

Sincerely,

Nadav Peleg

Dear Editor:

We would like to thank the editor's and all reviewers' valuable suggestions and comments on the manuscript. These comments have further improved the quality of the current manuscript. All point-by-point responses are presented in our replies and we have carefully revised the manuscript based on these comments.

Sincerely,

Wei Li and co-authors

**Reply to Referee comment 1**

**Emma Daniels (Referee)**

Great to see you have improved the paper substantially. My main worry that remains is the thresholds of extreme precipitation that you choose to investigate. The 99.95th percentile should not be used in my opinion because amount of daily data is way too small to trust such a statistic. Instead go with the 90th percentile for example. Something you can probably not implement in the current research but might want to remember for future work is the sub-tiling (mosaic) option for Noah.

Thanks for the comments and suggestions. We have replaced the 99.95th percentile summer rainfall with the 90th percentile summer rainfall, and some of the results are displayed below (Fig. 4, Fig. 9 and Fig. 10). In Fig.4 and Fig.9, the spatial distributions of the changes in 90th percentile summer rainfall are more similar to those in average summer rainfall than those in 99th percentile summer rainfall. In Fig. 10, the area average changes in 90th percentile summer rainfall have same regulations as those in 99th percentile summer rainfall with smaller variation ranges. Overall, replacing the 99.95th percentile summer rainfall with the 90th percentile summer rainfall makes no differences to the conclusions. For the sub-tiling (mosaic) option for Noah, we will take it into consideration in our future researches. Thanks for the suggestion.

[Figure]

**Figure 4. The bias of (a) average summer rainfall (%), (b) 90th percentile summer rainfall (%) and (c) 99th percentile**

**summer rainfall (%) between the 2010 scenario and observed data, and (d) the qq-plot of observed rainfall versus simulated rainfall. The stippling regions show statistically significance of bias identified by t-test at a 5% significance level.**

[Figure]

**Figure 9.** The changes in (a-b) average summer rainfall (mm), (c-d) 90th percentile summer rainfall (mm/day) and (e-f) 99th percentile summer rainfall (mm/day) between the 20% scenario and 2010 scenario, and between the 50% scenario and 2010 scenario. The stippling regions show statistically significance of changes identified by t-test at a 5% significance level.

[Figure]

**Figure 10.** The changes in (a) average summer rainfall (mm), (b) 90th percentile summer rainfall (mm/day) and (c) 99th percentile summer rainfall (mm/day) between the two hypothesis scenarios (20% and 50% scenarios) and 2010 scenario in ALL-YRB and PDG-YRB area. The blue boxes represent the 20% scenario, while the red boxes represent the 50% scenario.

**Reply to Referee comment 2**

**Anonymous Referee #2**

The authors addressed most of my previous comments. But after reading the revised manuscript, there are still some issues related to unclear methodology which requires clarifications.

Thanks for the comments. All point-by-point responses are presented as follows and we have carefully revised the manuscript based on these comments. For clarity, all comments are given in the original version, while responses are marked in blue.

I am confused by the simulation length. In L150: they wrote that "The simulated period was 11 years from 2000 to 2010"; in L162 they wrote that "The simulation length was 3 months from June to August". It is unclear to me is it a continuous simulation from Jan 2000 to Dec 2020, or just each JJA months from 2000 to 2020? For the former case, spin-up using the first year would be enough. For the latter case, it is effectively 11 independent experiments initialized from different years. It means the experiment needs to start earlier than June for spin up purpose (e.g., April or May).

Sorry for the confusion. In this study, we conducted a continuous simulation from Jan 2000 to Dec 2020, with the first year taken as spin-up time. In L162, the simulation length is the length of period we used to determine the most suitable parameterization schemes. As we have explained that the choice of parameterization schemes is based on the simulation of 2005 summer in L159, we have deleted the sentence in L162 "The simulation length was 3 months from June to August" to avoid redundancy and confusion.

L182: "The 99th percentile is the multiyear average value from the 99th percentile rainfall in each year". Please justify this choice. What if one uses the 99% percentile calculated from the entire study period. Will these two different treatments produce similar results?

The choice that the 99th percentile is the multiyear average value from the 99th percentile rainfall in each year is based on previous studies (Zhai et al., 2003; Pan et al., 2010). We also calculated the 99th percentile from the entire study period and find that these two different treatments produce similar results (Fig. R1).

References:

Zhai, P., and Pan, X.: Change in Extreme Temperature and Precipitation over Northern China During the Second Half of the 20th Century, Acta Geographica Sinica, 58, 1-10, https://doi.org/10.11821/xb20037s001, 2003. (in Chinese)

Pan, A., Fan, S., and Chen, H.: Characteristic of extreme climate change over Jiangsu Province in the last 45a, Scientia Meteor Sinica, 30, 87-92, https://doi.org/10.3969/j.issn.1009-0827.2010.01.014, 2010. (in Chinese)

[Figure]

**Figure R1. The bias of 90th and 99th percentile summer rainfall (%) between the 2010 scenario and observed data**
**calculated from the multiyear average (Fig. R1a and R1c); the bias of 90th and 99th percentile summer rainfall (%)**
**between the 2010 scenario and observed data calculated from the entire study period (Fig. R1b and R1d).**

Figure 4: both the absolute and relative biases are useful. Please provide both and keep one in main text and the other one in SI.

Thanks for the comments. We have added the absolute biases for Fig.4, and put it in the Appendix (Fig. S1).

[Figure]

**Figure S1. The bias of (a) average summer rainfall (mm), (b) 90th percentile summer rainfall (mm/day) and (c) 99th percentile summer rainfall (mm/day) between the 2010 scenario and observed data, and (d) the qq-plot of observed rainfall versus simulated rainfall. The stippling regions show statistically significance of bias identified by t-test at a 5% significance level.**

L203: what temperature variable from WRF was used to compare with what temperature variable from ERA-5 and what from observation? Is it all surface skin temperature or something else? There seems to be systematic bias in the temperature. The authors need to make sure they used the consistent and correct temperature variable for this comparison, as different temperature variables are not comparable.

The temperature variable used to validate the model performance was the 2m air temperature. We have checked that we used the consistent and correct temperature variable for this comparison. We also notice that the air temperature is systematically biased and similar results have been found in other studies. For example, Zhang et al. (2017) found that there was a cold bias in the eastern China when simulated by the WRF model, and the bias was < 5℃. Yan et al. (2021) also showed that the WRF model produced large cold bias over whole China expect for the northwestern Xinjiang. We have added the explanations in the manuscript.

References:

Zhang, X., Xiong, Z., Zhang, X., Shi, Y., Liu, J., Shao, Q., and Yan, X.: Simulation of the climatic effects of land use/land cover changes in eastern China using multi-model ensembles, Global and Planetary Change, 154, 1-9, https://doi.org/10.1016/j.gloplacha.2017.05.003, 2017.

Yan, Y., Tang, J., Wang, S., Niu, X., and Le, W.: Uncertainty of land surface model and land use data on WRF model simulations over China, Climate Dynamics, https://doi.org/10.1007/s00382-021-05778-w, 2021.

L225-226: I think the authors need also to tell us what land cover changes happened between 1990 and 2010. This helps to understand the cause of simulated precipitation change.

From Table 1, we can see that from 1990 to 2010, the area of cropland decreased from 29.15% to 28.48% for the whole basin, the area of forest increased from 42.82% to 43.60%, the area of grassland decreased slightly from 23.50% to 23.13%, the area of water and wetland increased slightly from 1.65% to 1.79%, the area of urban increased from 0.19% to 0.86%, and the unused land decreased from 2.69% to 2.14%. We have added the details of land cover changes between 1990 and 2010 in the section of Discussions.

L313: For "LHF increases by $2.08\times10^3$ and $4.82\times10^3$ W/m$^2$ for the 20% and 50% scenarios, respectively", I think there might be a mistake here. According to Fig 13, the largest LHF change is from -20 to 20 W/m$^2$, how it is possible that this value become two magnitude larger in the multiyear average? Same issue for sensible heat numbers.

Sorry for the confusion. We summed up the changes of LHF and SHF over the whole basin without dividing by the number of model grids in the basin, that was the reason why the changes were so large. After dividing by the number of model grids in the basin, the LHF increases by 0.26 and 0.61 W/m$^2$ for the 20% and 50% scenarios, respectively. Meanwhile, the SHF decreases by 0.54 W/m$^2$ and increases by 0.54 W/m$^2$ for the 20% and 50% scenarios, respectively. We have revised the descriptions in the manuscript.

L326: what is the temperature before this part?

The temperature before this part is the 2m air temperature. To keep consistent, we have replaced the surface skin temperature in this part with 2m air temperature. As the changes in 2m air temperature are almost the same as changes in surface skin temperature, it makes no difference to the conclusions.

**Reply to Referee comment 3**

**Anonymous Referee #3**

My comments on the manuscript "Impacts of land use/cover change and reforestation on summer rainfall for the Yangtze River Basin" have largely been handled. There are still English mistakes, but it doesn't affect the readability of text much. The paper needs further English editing to aid readability.

Thanks for the comments. The manuscript has been further proofread, and we have carefully checked the whole paper to
improve the readability.

I do believe that the paper would be more impactful if the authors made more of an attempt to understand the differences between the 20% forest cover scenario and the 50% forest cover scenario. Why would the 20% scenario produce larger changes than the 50% scenario? The surface fluxes look similar across the different scenarios, but the surface temperature and relative
humidity are quite different (and even opposite in sign). The authors allude to perhaps changes to horizontal wind, but this would be very interesting if they chose to discuss this further. The authors might look to Eiras-Barca et al., 2020 (doi: 10.1111/nyas.14364) for inspiration as this study looks at deforestation and assesses moisture transport and surface roughness changes as a result.

Thanks for the advices. We agree with the reviewer that deeper analyses of the differences between the 20% and 50% reforestation scenarios are necessary. After reading Eiras-Barca et al., 2020 (doi: 10.1111/nyas.14364) recommended by the reviewer and other references, we analysed the spatial changes in the wind at 10m and short-wave radiation to further explore the potential reasons leading to the differences between two reforestation scenarios. Fig. A7 shows that the 10m wind decreases in most places of the Yangtze River Basin for both scenarios, which is as expected because reforestation increases the surface
roughness. However, the 10m wind increases around the reforested areas, accelerating the moisture export from the forest. It is worth noting that areas with an increase in 10m wind are more expansive for the 50% scenario than for the 20% scenario, which means that more moisture is transported from the forest to other places for the 50% scenario. In addition, from the changes in wind direction in Fig. A7, moisture exported from the forest is transported towards the southern regions and finally flow out the Yangtze River Basin. The above analyses further prove that the differences between the 20% and 50%
reforestation scenarios are mainly caused by the changes of horizontal wind. Moreover, we notice that Eiras-Barca et al. (2020) used a water vapor traces tool to track the moisture. This tool can also be used in our future studies to better study the impacts of reforestation on moisture transportation. As for the short-wave radiation (Fig. S3), it has similar spatial changes with the surface skin temperature. The 2m air temperature decreases where the short-wave radiation decreases, further leading to the increases in 2m relative humidity and water vapor mixing ratio. We have added the analyses of 10m wind in the section of
Discussions and kept the spatial changes of short-wave radiation in the Appendix.

References:
Eiras-Barca, J., Dominguez, F., Yang, Z., Chug, D., Nieto, R., Gimeno, L., and Miguez-Macho, G.: Changes in South
American hydroclimate under projected Amazonian deforestation, Ann N Y Acad Sci, 1472, 104-122, 10.1111/nyas.14364,
2020.

[Figure]

**Figure A7. The changes in (a-b) 10m wind (m/s) between the 20% scenario and 2010 scenario, and between the 50%
scenario and 2010 scenario. The stippling regions show statistically significance of changes identified by t-test at a 5%
significance level.**

[Figure]

**Figure S3. The changes in (a-b) short wave radiation (W/m$^2$) between the 20% scenario and 2010 scenario, and between
the 50% scenario and 2010 scenario. The stippling regions show statistically significance of changes identified by t-test
at a 5% significance level.**